# Molecular elucidation of cement hydration inhibition by silane coupling agents

Binmeng Chen[1], Meng Wang[1], Hegoi Manzano ®[2], Yuyang Zhao[1] & Yunjian Li ®[3] ✉

Silane coupling agents are widely recognized to retard early hydration when incorporated into fresh cement paste, yet the atomic-level mechanisms underlying their effects on clinker dissolution, such as adsorption of silane monomer onto reactive surface sites and modification of ion detachment pathways, remain unexplored. Here we show dissolution behavior of tricalcium silicate ($Ca_3SiO_5$) under 3-aminopropyl triethoxysilane impact using ab initio metadynamics, with experimental validation of the retardation effects in silane-treated pastes. The shielding effect of silane induces shifts in free energy changes of stepwise calcium dissolution from negative to positive and alters the most stable Ca coordination state during dissolution, resulting in the transition of dissolution from spontaneous to non-spontaneous. Specifically, hydrolyzed silane adsorbs dissociatively onto the $Ca_3SiO_5$ surface by forming ionic Ca-O bonds, thereby occupying reactive sites and introducing steric hindrance. This, in turn, impedes coordination interactions between calcium ions and water molecules. Experimental results further corroborate these interactions, as evidenced by reduced calcium concentrations in silane-treated pastes, which in turn slowed the hydration process. These findings offer nanoscale insights into the role of SCAs in cement hydration and provide a foundation for future research into the complex interactions within organic/cement systems.

The demand for higher-durability concrete that can withstand ion corrosion and dissolution degradation is increasing, particularly in harsh service environments such as marine settings. In these environments, enhancing the surface hydrophobicity of concrete is crucial for protecting the underlying substrate from water infiltration[1]. One effective strategy to improve the impermeability of concrete is the incorporation of silane coupling agents (SCAs)[2], SCAs typically have a bifunctional structure consisting of organic and inorganic components[3], which can be expressed as $(RO)_3 - Si - X$. In acidic or alkaline solutions, the alkoxy groups$(-RO)_3$, such as $-CH_3$ or $-C_2H_5$, undergo hydrolysis to form hydroxyl groups ($-OH$)[4]. These hydroxyl groups can then participate in condensation reactions with calcium silicate hydrate (C-S-H), forming robust siloxane linkages on particle surfaces[5]. The organo-functional groups provided by SCAs impart hydrophobicity to concrete, effectively preventing water from penetrating its pores[6].

In practical engineering, the application of silane in concrete can be categorized into two approaches: surface treatment and initial admixture[7]. In surface treatment, the hydrolyzed silane monomers diffuse into the concrete through capillary pores, driven by a concentration gradient[8]. These monomers subsequently adsorb physically or bond chemically with C-S-H phases, gradually developing interfacial adhesion[9]. Alternatively, in the initial admixture method, pre-treated silane is directly incorporated into the fresh paste, ensuring a more uniform distribution of the silane[10]. The highly alkaline environment provided by cement hydration facilitates the endothermic hydrolysis

[1]Institute of Applied Physics and Materials Engineering, University of Macau, Avenida da Universidade, Taipa, Macao SAR, China. [2]Physics Department, Faculty of Science and Technology, University of Basque Country UPV/EHU, Barrio Sarriena s/n, 48940 Leioa, Spain. [3]Faculty of Innovation Engineering, Macau University of Science and Technology, Avenida Wai Long, Taipa 999078 Macao SAR, China. ✉e-mail: liyunjian@must.edu.mo

of silane, leading to C-S-H condensation and the formation of a Si-O-Si network[11]. Therefore, blending SCAs in fresh paste is regarded as a more effective approach for designing durable cementitious materials, while also enhancing the workability of fresh paste. A study noted that incorporation of 3-aminopropyl triethoxysilane (APTES) into the montmorillonite partial substituted system significantly improved the fluidity of the cement paste, increasing it from 26.89% to 65.25%, and induced the formation of acicular hydration products that improved microstructure[12]. Recent reports have also confirmed that silane modification improves the flexural performance of cement paste[13]. Feng et al. found that incorporating a small amount of silanes increased both the flexural and compressive strength of ordinary mortars after 7 and 28 days of curing[14].

Despite these benefits, the inclusion of SCAs can introduce adverse effects, such as the inhibition of initial cement hydration. Rodríguez et al.[15] reported that incorporating SCAs significantly delays the emergence of hydration peak, thereby inhibiting initial hydration. This effect could be attributed to the increased repulsive force between free water and cement particles, introduced by the hydrophobic groups of the silane. Kong et al.[16] also identified this inhibitory effect on cement hydration and indicated that the influence of the hydrolysis byproducts, such as alcohols is negligible. Moreover, the hydrophobic groups of silanes negatively impact the silicate polymerization and hydrogen bond formation, thereby influencing the nucleation of the hydration product C-S-H (calcium-silicate-hydrate) and further retarding the hydration. Xie et al.[17] demonstrated that incorporating silane reduces the quantity and cumulative density of C-S-H, subsequently hindering the cement hydration process. More recent findings indicated that silane hydrolysates can form complexes with calcium ions, potentially accounting for the delayed hydration[18]. Despite these insights, there remain gaps in understanding the mechanisms of silane-induced delayed hydration. While the two aforementioned hypotheses provide plausible explanations, hydration is an inherently complex and prolonged process, where silane may exhibit different retarding mechanisms at various stages. It is worth mentioning that the mechanism of complexation and inhibition of C-S-H growth appears more relevant to the post-induction stage of hydration. However, the precise impact of silane on clinker dissolution during the induction stage warrants further exploration. Given that mineral dissolution occurs instantaneously (within seconds), direct observation poses significant challenges. As such, atomic-scale simulations offer a promising approach for studying this phenomenon.

Atomistic simulations have emerged as an alternative method to provide valuable insights into the nanoscale mechanisms of the inhibition of SCAs on cement hydration[19–21]. Molecular dynamics (MD) simulations using empirical force fields have been applied to explore lots of physical properties of cement-based materials, such as the efficiency of solution transport in calcium-silicate-hydrate (C-S-H) nanopores functionalized with silane[22]. Additionally, MD simulations based on reactive force field (ReaxFF) are extensively used to model clinker hydration processes. Li et al.[23] demonstrated that the degree of hydrolytic separation on the surfaces of M3- and T1-C3S is comparable and independent of the polycrystalline state of the crystal. Xu et al.[24] found that the complete dissolution of calcium ions in C₃S required two dissolution processes. Otherwise, density functional theory (DFT) calculations have also been performed to elucidate the adsorption behavior of water on clinker phases and SCAs on the C-S-H surface without considering the dynamics information of the reaction[25]. As theoretical methods continue to advance, ab initio molecular dynamics (AIMD) simulations have been increasingly adopted to accurately probe chemical reactions involving electron transfer, including key transformations in cement hydration such as adsorption and nucleation[26]. In this context, enhanced sampling techniques like metadynamics (MetaD), offer a powerful approach for exploring significant configurational changes within a system, from reactants to

products[27]. Recently, it has been reported that the combination of AIMD and MetaD successfully drew a full picture of the calcium dissolution from the $Ca_3SiO_5$ and $Ca_2SiO_4$ surface[28,29], which sheds light on the understanding of the effect of SCAs on $Ca_3SiO_5$ dissolution.

To elucidate the retardation effect of SCAs on cement hydration, we employed a combination of ab initio simulations and experimental corroboration to investigate the impact of APTES on the early $Ca_3SiO_5$ hydration at the nanoscale. First, the accuracy of the MetaD sampling model was validated through DFT adsorption tests, followed by the exploration of the dissolution behavior of surface penta-coordinated calcium under the influence of APTES using ab initio MetaD simulations. The results indicated that the shielding effect of SCAs causes the dissolution of calcium ions to shift from a spontaneous to a non-spontaneous process. Secondly, the early hydration kinetics of silane-treated cement paste were examined through hydration experiments, which not only corroborated the interactions between $Ca_3SiO_5$ particles and APTES molecules but also demonstrated that the reduced concentration of calcium ions in the hydrolytic solution is a critical factor in the retardation of cement hydration. Finally, these discoveries provide a molecular-level explanation for the delayed hydration effect of SCAs, contributing to a deeper understanding of the initial hydration of cement in organic systems.

## Results

### Adsorption behaviors of APTES/Ca₃SiO₅ interface

Interactions between APTES and the $Ca_3SiO_5$ surface were studied by DFT calculations on the adsorption of APTES on different reactive sites of the (111) $Ca_3SiO_5$ surface. When APTES comes into contact with the surface, there will be two types of adsorption modes on the $Ca_3SiO_5$ surface: molecular adsorption (MA) and dissociative adsorption (DA)[30]. For the MA adsorption, the -Si(OH)₃ group adsorbed on the surface, where the oxygen ion from APTES ($O_a$) formed an ionic bond with surface calcium ($Ca_s$), and the hydrogen ion from APTES ($H_a$) established a hydrogen bond with the surface oxygen ion($O_s$), as illustrated in Fig. 1a. The formation of these chemical bonds corresponds to the chemisorption. For the DS adsorption shown in Fig. 1b, one silanol group (Si-OH) in APTES was deprotonated, and the liberated proton was transferred to the free oxygen atom on the surface to form a hydroxide ion (OH⁻). The deprotonated oxygen in APTES adsorbed on the bridge sites of two $Ca_s$ ions forming two $Ca_s$-O bonds. Simultaneously, another two silanol groups in APTES also chemically bonded with the calcium ion.

Conclusively, Ca-O ionic bonds were formed in both systems, with the -SiOH group occupying adsorption sites near the calcium surface. Notably, the proton transfer that occured in the DA process implies stronger chemisorption, with more adsorption sites occupied, and might impede the adsorption of water molecules nearby in solution environments. The formation of hydrogen bonds between the silanol group and Si tetrahedra on the $Ca_3SiO_5$ surface in MA might offer chemical potential for the formation of Si-O-Si linkages.

The adsorption energy ($E_{ads}$) was calculated by Supplementary Equation (1) to evaluate the stability of the two modes[31] (Supplementary Note 2), as shown in Fig. 2b. The adsorption energy of the MA adsorption was −126.85 kJ/mol, indicating that the total energy of the system decreases, and the structure became more stable following adsorption. In contrast, the adsorption energy for the DA adsorption was approximately −263.11 kJ/mol, which is greater than MA adsorption. This gap was associated with the increased proton transfer and formation of more Ca-O bonds. Consequently, the DA adsorption mode was significantly more stable, making it more likely to take place when APTES interacts with the $Ca_3SiO_5$ surface. Since the adsorption test is conducted in a vacuum, it may overestimate the adsorption strength of silane in solution. However, when compared to the previously reported dissociative adsorption energy of water molecules in a vacuum environment[32,33], APTES exhibited significantly higher

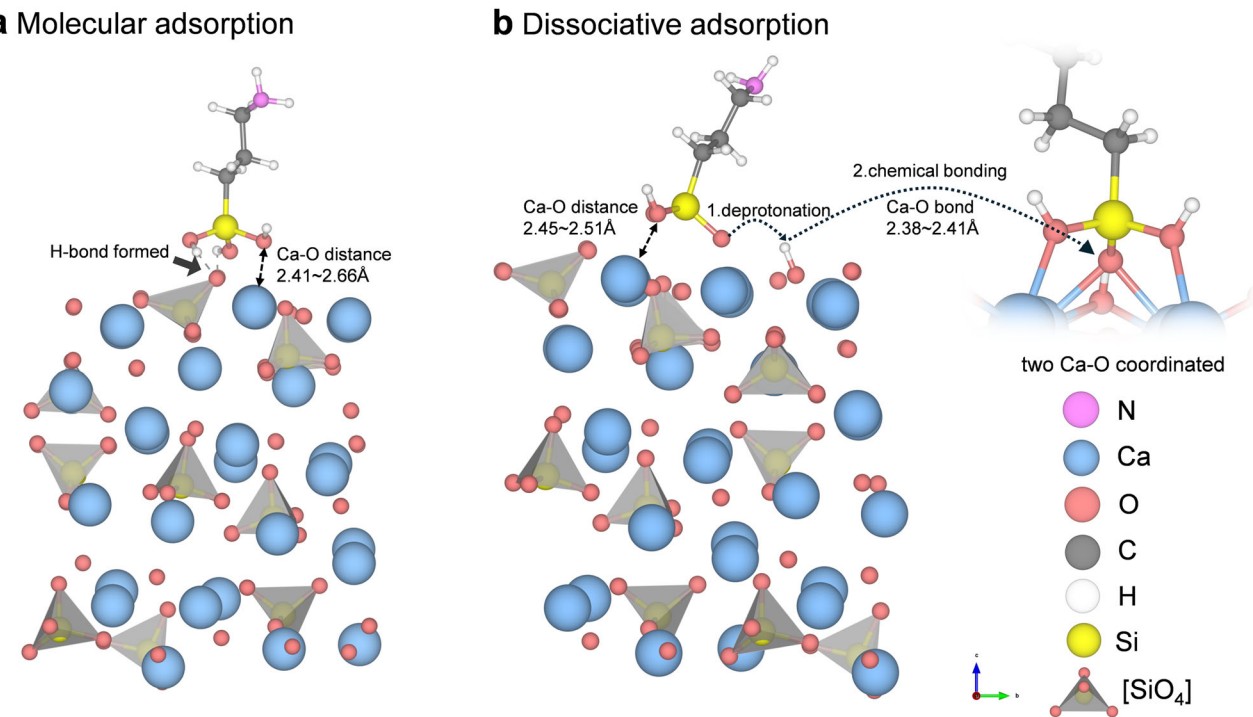

**Fig. 1 | Adsorption mode of APTES on the Ca₃SiO₅ surface. a** Molecular adsorption. **b** Dissociative adsorption. The lengths of the generated Ca-O ionic bonds are indicated.

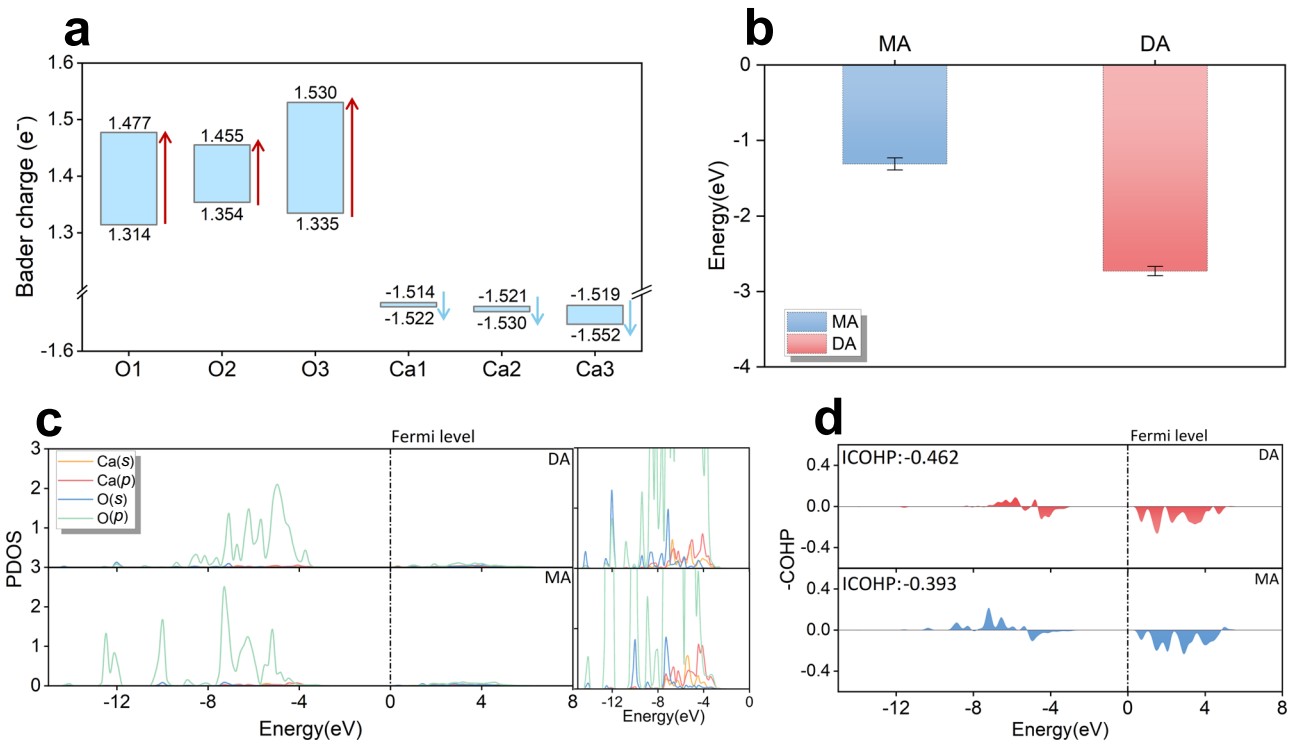

**Fig. 2 | Electron analyses of the DA and MA adsorption. a** Bader charge transfer pre- and post-adsorption. **b** Adsorption energies of molecular adsorption and dissociative adsorption, error bars reflect the standard deviation (s.d.) in the adsorption energy values. **c** Partial density of states between surface calcium and APTES oxygen, with enlarged parts of the corresponding orbitals shown on the right. The s and p in the brackets in the legend define the s and p orbitals of the atom. **d** Corresponding crystal orbital Hamilton populations between surface calcium and APTES oxygen. The Fermi levels have been calibrated to 0 eV.

values, indicating its greater chemical reactivity with the Ca₃SiO₅ surface.

Charge transfer during the chemisorption of APTES molecule on the Ca₃SiO₅ surface was analyzed by Bader charges before and after adsorption[34]. Results are presented in Fig. 2a, in which O1, O2, Ca1, and Ca2 represent APTES oxygen atoms and surface Ca atoms forming Ca-OH bonds in MA and DA systems, respectively, O3 and Ca3 correspond to APTES oxygen atom and surface Ca atom that forming Ca-O bonds

after deprotonation in DA system, respectively. Upon forming Ca-OH bonds, O1 and O2 gained 0.163 $e^-$, and 0.102 $e^-$, respectively, while Ca1 and Ca2 lost 0.008 $e^-$ and 0.009 $e^-$, suggesting that the resulting ionic bonds are relatively weak. In contrast, after deprotonation and the formation of Ca-O bonds, O3 gained 0.195 $e^-$ and Ca3 lost 0.33 $e^-$, more charge transfer verifies the formation of a stronger ionic bond[35].

Crystal orbital Hamilton populations (COHPs) and density of states (DOSs) of bonded atoms (Ca$_s$-O$_a$) were calculated to verify chemical bond existence[36–38]. A detailed look at the PDOS diagram (Fig. 2c) showed a slight overlap between calcium and oxygen orbitals in the occupied states near the Fermi level, corresponding to the closed-shell interactions characteristic of ionic bonds. The associated COHP analysis indicated a clear bonding contribution in the range of (−8) to (−4) eV near the Fermi level. Additionally, we calculated the correlative -ICOHP values of the Ca-O$_a$ bonds under dissociative and molecular adsorption conditions to assess the bond strength. These values were obtained by integrating the COHP below the Fermi level. In the DA process, the O$_a$ atom forms double Ca-O$_a$ bonds with a -ICOHP value of 0.462, compared to 0.393 in the MA process, indicating stronger interaction (Fig. 2d). Based on proton transfer and chemical bond formation, it can be concluded that the DA system involves chemisorption. Additionally, analysis of the electronic structures further supported these findings, confirming the formation of the Ca-O bond in both MA and DA adsorption systems. This validates the rationale behind the selection of CVs in the sampling process.

## Ca ion dissolution pathways under the influence of APTES

The negative adsorption energy of APTES on the Ca$_3$SiO$_5$ surface suggests that APTES preferentially adsorbs at reactive sites, potentially influencing the dissolution of Ca$_3$SiO$_5$. To investigate these atomic-level interactions, ab initio well-tempered metadynamics (WT-MetaD) simulations were performed to examine the impact of APTES on the dissolution pathway of calcium ions. The two-dimensional free energy

surface (FES) depicted in Fig. 3b illustrateed the free energy barriers along the collective variables (CVs), with the corresponding numerical values represented by the color gradient. The free energy landscape for calcium ions revealed a range of minima distributed over a checkered pattern spanning 120 kJ/mol. These minima were associated with a reduction in the number of bonds to surface oxygen atoms of Ca$_3$SiO$_5$ and an increase in bonds with oxygen atoms from APTES. As the simulations progress, the free energy surface revealed an addition/elimination mechanism akin to the hydrolysis process observed in the aluminum phase of Al$_2$O$_3$[27]. Notably, even when the coordination number of water molecules and calcium ions was included as a set variable, the bond between the oxygen atoms (O$_w$) of water molecules and calcium ions had not been observed during the sampling process with the addition of Gaussian peaks (Fig. 3a).

This absence was attributed to both the hydrophobic effect of APTES and the silicon group adsorbed near calcium ions, which occupied most of the available adsorption sites, thereby preventing coordination between water molecules and calcium ions. Furthermore, the pathway from calcium ion dissolution to the 5-coordinated state—identified as the most favorable reaction path in terms of kinetics and thermodynamics—was analyzed (Fig. 3c, d). Each step in the coordination transformation of calcium ions requires crossing two sequential energy barriers. For the initial state A, the process involveed breaking two Ca-O$_s$ bonds from the surface, with free energy barriers of ΔA‡(A-B) = 4.64 kJ/mol and ΔA‡(B-C) = 8.31 kJ/mol. Subsequently, the calcium ion coordinateed with an oxygen atom on the APTES silicon group to form state D, followed by the cleavage of another Ca-O$_s$ bond to reach state E. The free energy barriers for these steps were ΔA‡(C-D) = 17.32 kJ/mol and ΔA‡(D-E) = 19.97 kJ/mol.

It is important to emphasize that the final state of the WT-MetaD simulation aligns with the initial state (5,1), as this configuration exhibits the lowest energy and is thus the most stable on the free energy surface. As the coordination number decreases, the relative energies of

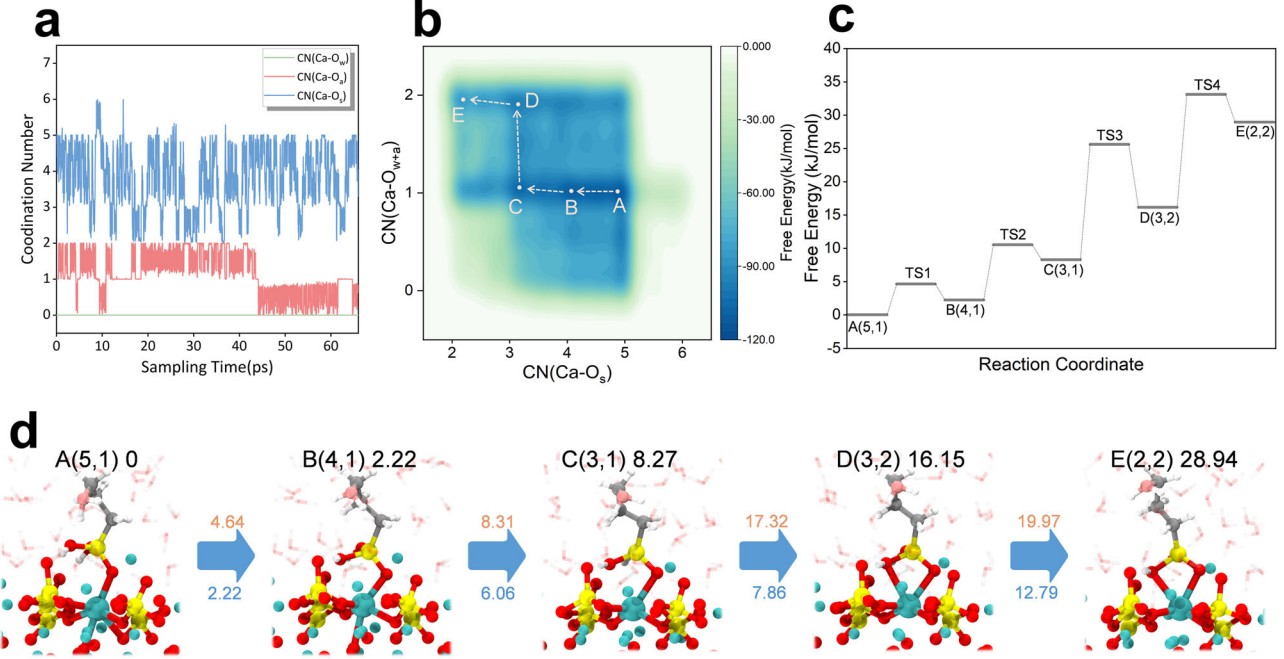

**Fig. 3 | Mechanisms of Ca$_3$SiO$_5$ dissolution under the influence of APTES.**
**a** Time-evolution of collective variables during the WT-MetaD simulation. **b** The two-dimensional free energy surface with variables of CN(Ca-O$_s$) and CN(Ca-O$_{w+a}$) for the APTES-containing system. **c** The reaction coordinate represents the dissolution process, with the distinct transition states labeled as TSn ($n$ = 1, 2, 3, 4...). **d** Corresponding snapshots illustrating the configuration evolution along the reaction pathway. The state numbers, coordinates on the FES, and the Helmholtz free energy values (in kJ/mol) relative to state A are indicated in the upper right corner. The red values upon the blue arrow denote the free energy barriers (in kJ/mol), while the blue values beneath the arrows indicate the overall free energy changes between successive states (in kJ/mol). Atom color-coding is as follows: red for oxygen, yellow for silicon, white for hydrogen, cyan for calcium, gray for carbon, and pink for nitrogen. For clarity, water molecules are depicted in a transparent stick representation.

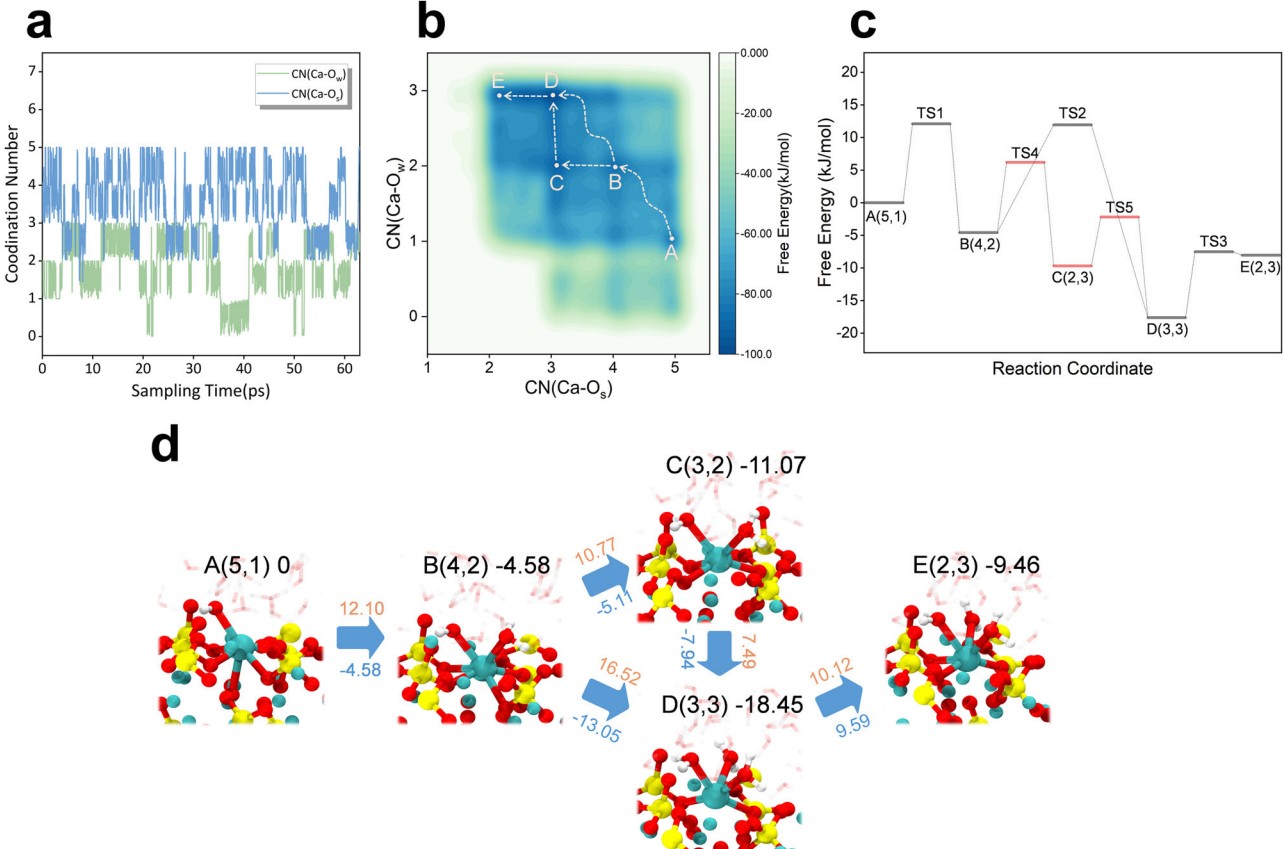

**Fig. 4 | Mechanisms of Ca₃SiO₅ dissolution in aqueous solution. a** Time-evolution of collective variables during the WT-MetaD simulation. **b** The two-dimensional free energy surface with variables of CN(Ca-O$_s$) and CN(Ca-O$_w$). **c** The reaction coordinate represents the dissolution process, with the distinct transition states labeled as TSn ($n$ = 1, 2, 3, 4…). **d** Corresponding snapshots illustrating the configuration evolution along the reaction pathway. The state numbers, coordinates on the FES, and the Helmholtz free energy values (in kJ/mol) relative to state A are indicated in the upper right corner. The red values upon the blue arrow denote the free energy barriers (in kJ/mol), while the blue values beneath the arrows indicate the overall free energy changes between successive states (in kJ/mol).

states B, C, D, and E progressively increase compared to state A. This observation suggests that in the presence of APTES, the calcium ion's stable state is more inclined toward a surface-bound position.

## Reactivity of the Ca₃SiO₅/water interface

The dissolution reactivity of the Ca₃SiO₅/water interface was investigated, and the corresponding free energy profile is presented in Fig. 4b. Obviously, the dissolution process of FES separated by Ca, which is completely exposed to the solution, differs from FES adsorbed by APTES. The time-evolution of CVs (Fig. 4a) demonstrates that the highest tri-coordination with water is observed on the current free energy surface. However, both processes initiate from the same stable state, A(5,1). Depending on the sequence, three potential reaction pathways existed from state A to the adsorption of a water molecule, followed by the breaking of a Ca-O$_s$ bond, leading to state B. From a thermodynamic perspective, it was more probable that these two steps occur simultaneously, as state B (4,2) exhibits lower energy than both states (4,1) and (5,2), with the corresponding free energy barriers for this process being ΔA‡(A-B) = 12.10 kJ/mol. Subsequently, along the reaction coordinate in Fig. 4c, Ca continued to adsorb a water molecule and broke a Ca-O$_s$ bond to reach the most stable C(3,3) state, which also represented the final state observed in MetaD simulations. For this step, there are two possible pathways. From a kinetic standpoint, the reaction is more likely to proceed along the B → C → D pathway (Fig. 4d), as the free energy barriers for this path−ΔA‡(B-C) = 10.77 kJ/mol and ΔA‡(C-D) = 7.49 kJ/mol−are lower than those of the B → D path (ΔA‡(B-D) = 16.52 kJ/mol). To

proceed to the E(2,3) state, Ca must overcome two consecutive free energy barriers of 10.12 kJ/mol.

In contrast to calcium ions that are fully exposed in an aqueous solution, the dissociative adsorption of APTES reduces the free energy barrier associated with initially breaking the two Ca-O$_s$ bonds, thereby promoting the kinetic process. However, APTES inhibits the interaction between calcium ions and nearby water molecules, resulting in a consistent CN(Ca-O$_w$) value of zero. From an energetic standpoint, the stable state of calcium is altered under the influence of silane, which leads to penta-coordination with the surface (CN(Ca-O$_s$) = 5). This is in contrast to the tri-coordinated calcium in an aqueous solution (CN(Ca-O$_s$) = 3), suggesting that calcium is more likely to stabilize at the surface following silane treatment. Additionally, the change in the free energy of the dissolution reaction further suggests that the spontaneity of the Ca dissolution process is influenced by silane. The transition from a decreasing to an increasing free energy change indicates that the dissolution reaction shifts from spontaneous to non-spontaneous, which provides a fundamental explanation for the inhibition of calcium dissolution by silane.

## Dissolution inhibition mechanism governed by steric hindrance

In order to fully elucidate the inhibition mechanism, a detailed analysis of the interatomic interactions in the final state is necessary. A 10 ps AIMD simulation was performed for the final state on the free energy surface to analyze its detailed kinetics behaviors. Based on the density distribution curve of water molecules, the steric hindrance effect of APTES molecules caused the stratification of water molecules near the

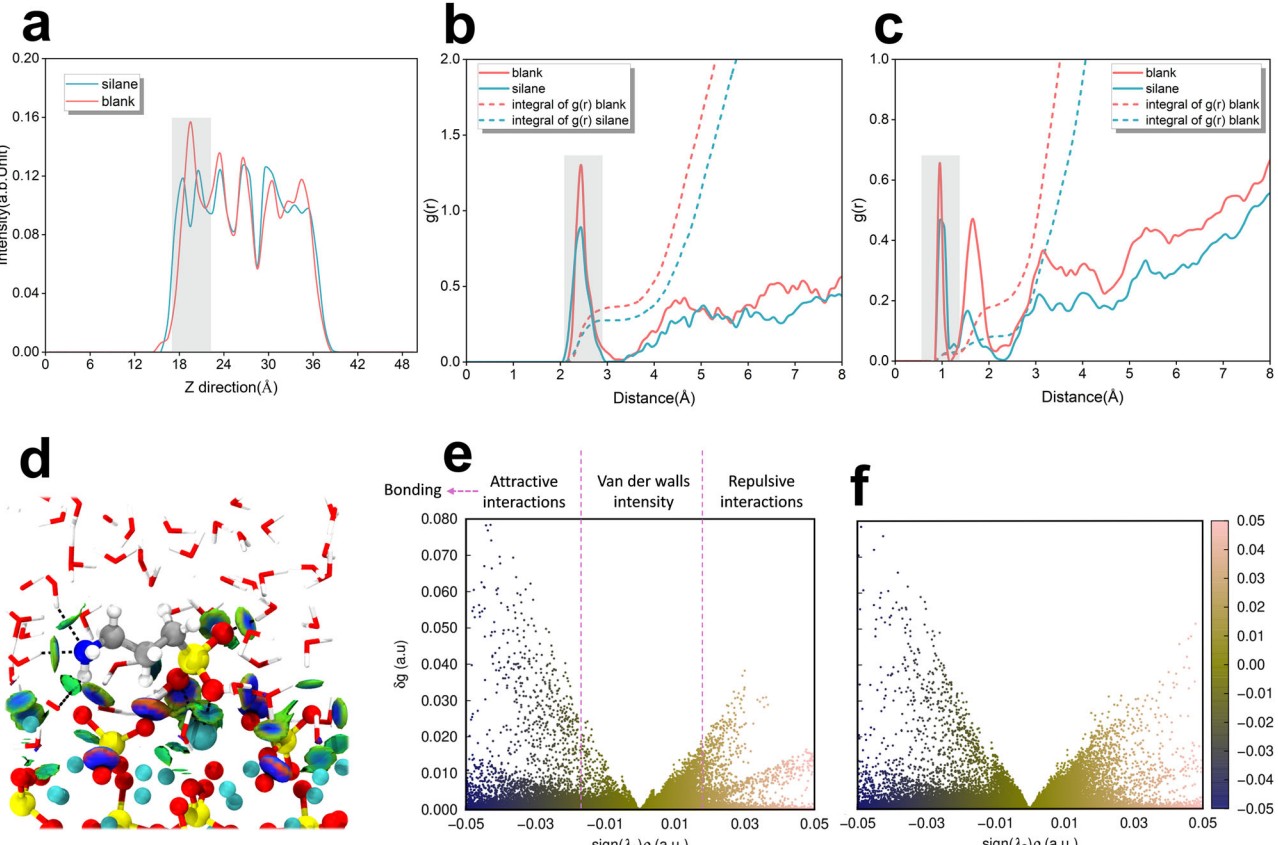

**Fig. 5 | Structural and interaction analysis of the final states in two distinct dissolution environments. a** The one-dimensional density profile of water molecules along the Z-axis. Gray shading marks the areas where water molecules are layered. **b** the pair RDF between calcium ions and water oxygen atoms. **c** the RDF between slab oxygen atoms and water hydrogen atoms. The gray shading marks the first coordination peak. Dotted lines in panels **b** and **c** represent the integration curves of the RDFs. The ordinate values at the intersection points between the first RDF valley and integrals indicate the average coordination numbers for the respective pairs in the system. **d** The atomic isosurfaces of the Independent gradient model based on Hirshfeld (IGMH) for the APTES-containing system in a (3,3) coordination state. Blue and green isosurfaces between atomic pairs indicate attractive interactions, with strengths comparable to hydrogen bonds or van der Waals forces. In contrast, the red isosurfaces represent repulsive interactions, such as steric hindrance, between atoms. **e** The quantitative IGMH scatter plot of blank system, highlighting the range of weak interaction intensity as indicated by the $\text{sign}(\lambda_2)\rho$[39]. **f** The IGMH scatter plot of the APTES system.

surface, suggesting that their diffusion is restricted. The RDFs for the $Ca\text{-}O_w$ and $O_s\text{-}H_w$ pairs in the system were calculated using Supplementary Equation (2) to analyze the adsorption and bonding behaviors of water molecules on the $Ca_3SiO_5$ surface (Supplementary Note 2), as illustrated in Fig. 5b, c.

In the APTES-containing system, the first peaks of the radial distribution function (RDF) curves for both types of pairs were significantly reduced. This reduction was attributed to the lateral adsorption of the APTES oxygen near calcium atoms, which occupied surface sites and, through steric hindrance, reducing the amount of water adsorbed on the surface. Additionally, the corresponding integral curves were obtained by integrating the RDF curves, where the ordinate value at the intersection of the first peak and valley of the RDF represents the average number of coordination bonds for the respective pairs in the system. A detailed analysis showed that the total number of $Ca\text{-}O_w$ bonds decreased markedly, while the decrease in $O_s\text{-}H_w$ bonds is relatively minor. This indicated that the steric hindrance effect of APTES predominantly restricts the chemisorption and bonding between calcium and water oxygen, thereby inhibiting further dissolution from the surface.

To gain a deeper understanding of the steric effects of the APTES molecule, the Independent Gradient Model for Hirshfeld partitioning (IGMH) analysis was performed using the Multiwfn package based on the electronic structures[39]. The weak interactions among the $Ca_3SiO_5$ slab, water molecules, and the APTES molecule are present in Fig. 5d.

Obviously, monitored calcium exhibited robust attractive interactions with APTES oxygen, and this should be the Coulomb interactions that originated from the charge difference. Strong hydrogen bonds were observed between the $\text{-}SiO_3$ group and the silica tetrahedron as well as water molecules, with the blue equivalent face representing the attractive region. This suggested that proton exchange is highly dynamic, corresponding to the RDF curves. Additionally, hydrogen bonding was identified between the amino group, water molecules, and ionic oxygen on the surface, with a strength comparable to van der Waals interactions, which explained the lateral adsorption of APTES molecules on the surface. Analysis of the scatter plot (Fig. 5e, f) revealed that, compared to the blank system, the APTES-containing system dispersed more scatters in the repulsive region and fewer in the attractive region, indicating that the addition of APTES induces a significant steric hindrance effect, which inhibited the chemisorption of water molecules on the surface.

## Experimental corroboration on APTES-induced inhibition of $Ca_3SiO_5$ hydration

Hydration kinetics of $Ca_3SiO_5$ under conditions of DI and APTES solution were analyzed by hydration heat tests. The Ca dissolution process dominates the initial rapid dissolution period and partially the induction and accelerating period. As shown in Fig. 6, the induction period and hydration peak of the blank group occurred always earlier than that of the silane group, and the induction period was

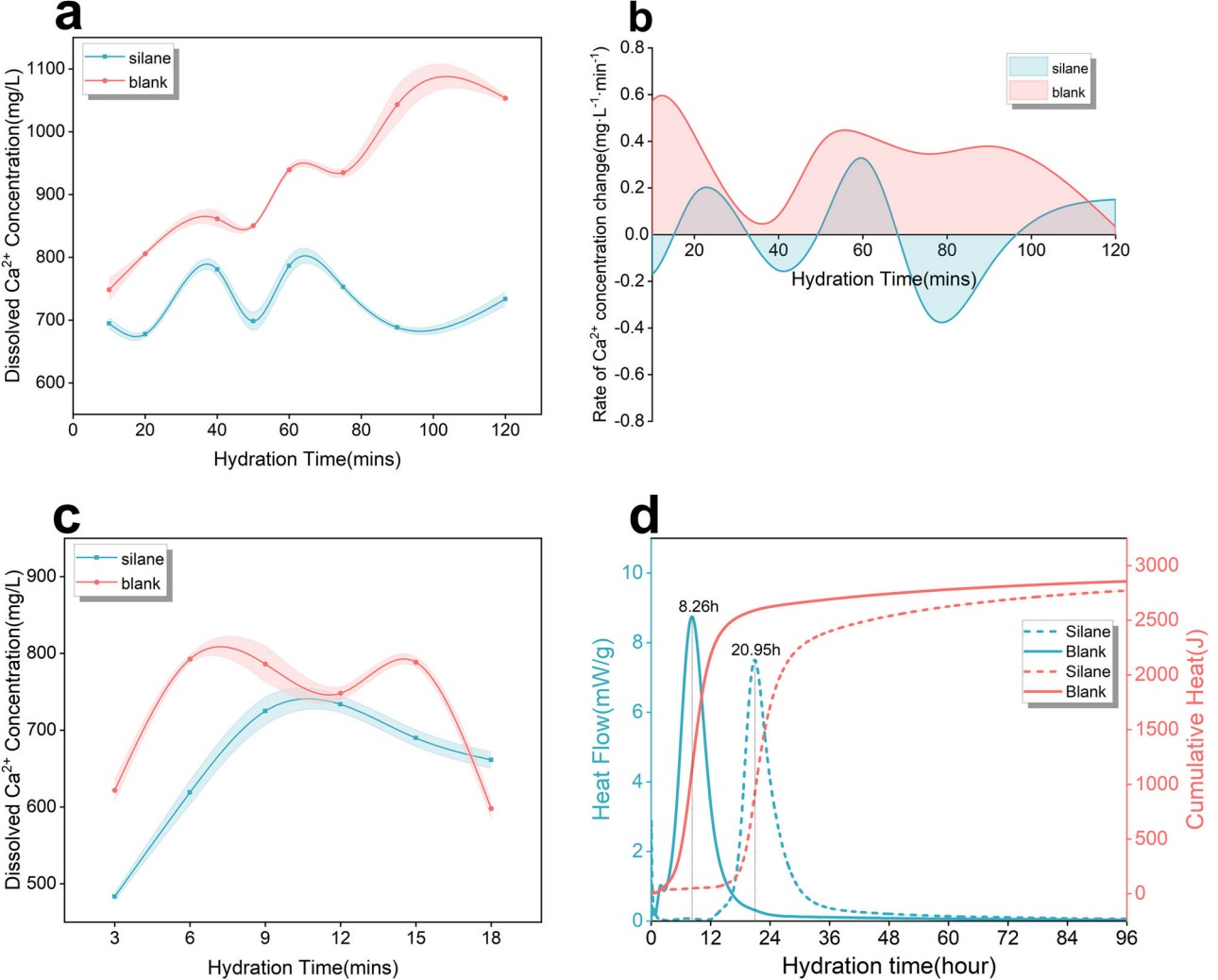

**Fig. 6 | Quantitative analysis of the dissolution behavior of Ca₃SiO₅ treated with APTES. a** Concentrations of dissolved calcium ions in solution, depicted with error bands alongside the concentration curve. **b** Rate of change in dissolved calcium ion concentration, obtained through differentiation of the time-based curve. **c** Concentration profile of dissolved calcium ions over the initial 20-minute hydration period. **d** Heat flow and cumulative heat release curves of Ca₃SiO₅ during the hydration process. Error bars of **b**, **c** reflect the s.d. in the concentration values.

significantly prolonged in the silane-treated paste. Besides, the total heat release tended to decrease when APTES was added to the system. These results showed that there is an inhibition effect of APTES on the Ca₃SiO₅ hydration, especially on the Ca dissolution. Notably, the total heat release of the silane-treatment group was slightly lower than in the blank group, indicating that silane slightly retard the hydration degree of Ca₃SiO₅.

To further understand the impact of APTES on Ca₃SiO₅ dissolution, concentrations of dissolved calcium ions for groups with and without the addition of APTES were monitored along the hydration time by the inductively coupled plasma-optical emission spectrometry (ICP-OES) test. As shown in Fig. 6a, the concentration of dissolved calcium ions in the blank group was higher than that in the silane-treated paste at all sampling times. Additionally, time derivatives of the concentrations for these groups were calculated as illustrated in Fig. 6b. The rate of the change of concentration for the blank group was positive, suggesting that calcium dissolution predominates the hydration process during the sampling time, while the rate change of the concentration for the silane-treated group converted between positive and negative values, indicating calcium ions underwent a cyclical process of alternating between dissolution-dominated and consumption-dominated phases. Calcium ion concentration within

20 min by ICP analysis was taken at 3, 6, 9, 12, 15, and 18 min, as illustrated in Fig. 6c. The results revealed that additional silane significantly reduced both the dissolution and consumption rate of calcium ions. Notably, the calcium ion concentration in the silane-treatment group was substantially lower than that in the blank group during the initial 15 min of hydration. Therefore, based on this evidence, APTES indeed has an inhibition effect on the Ca dissolution, which is in line with the above simulation results.

X-ray photoelectron spectroscopy (XPS) analysis was further performed to study whether the inhibition effect of APTES on the Ca₃SiO₅ dissolution is derived from their interfacial interaction and the relevant point data could be obtained from Supplementary Table 1. The XPS spectra, fitted with a 2.5 eV full width at half maximum (FWHM) for the Ca2p and Si2p peaks, are displayed in Fig. 7. For the Ca₃SiO₅ samples hydrated in deionized water, the Ca2p and Si2p peaks remained at consistent binding energies of 346.7 and 101.0 eV, respectively, across four distinct hydration intervals. These energies were marginally lower than those reported by Rheinheimer (346.7 eV for Ca2p3/2 and 102.4 eV for Si2p following 12 min of hydration)[40]. A closer examination showed a broadening of the Ca2p and Si2p peaks after 2 h of hydration, corresponding to the formation of the disordered hydration product, calcium silicate hydrate (C-S-H)[41].

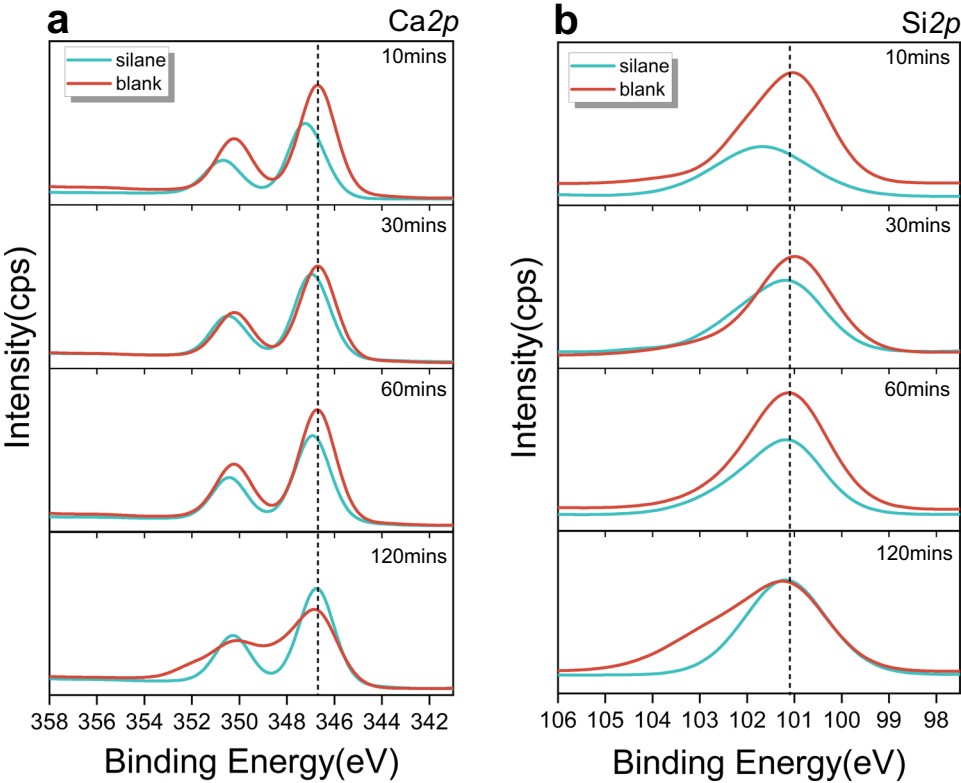

**Fig. 7 | XPS spectra at different hydration times have been treated by charge correction and deconvolution. a** XPS spectral curve of Ca$2p$. **b** XPS spectral curves of Si$2p$. Hydration time is indicated in the upper right corner.

Interestingly, XPS analysis indicated a shift in the binding energies after 10 minutes of hydration in a silane solution, with the Ca$2p$ peak moving from 346.7 to 347.2 eV and the Si$2p$ peak shifting from 101 to 101.2 eV. This shift indicated the bonding of APTES to the surface of Ca$_3$SiO$_5$ particles during the initial mixing, verifying the rationality of the computational model of MetaD simulation on Ca dissolution with the adsorption of APTES on the reactive site. As hydration progresses, the Ca$2p$ and Si$2p$ peaks return to their original positions, aligning with the unmodified peaks, indicating that these bonds are disrupted by the ongoing hydration reaction. Moreover, the lack of peak broadening for Ca$2p$ and Si$2p$ in the silane-treated samples after 2 h of hydration suggested that the presence of APTES may inhibit the nucleation and growth of C-S-H.

To elucidate the phase transformations during the hydration of Ca$_3$SiO$_5$, quantitative X-ray diffraction (QXRD) was conducted to assess the consumed Ca$_3$SiO$_5$ content while transmission electron microscopy (TEM) was carried out to monitor the microstructure evolution of the hydration product (Fig. 8a, b). QXRD curve showed that the characteristic peak of calcium hydroxide disappeared, even after 120 minutes of hydration corresponding to the accelerated dissolution stage. During this stage, the ion concentration in the solution had not yet reached the saturation threshold necessary for precipitation. Furthermore, phase components at each hydration interval of both groups were quantitatively calculated using fitting analysis to compare the mass content of Ca$_3$SiO$_5$ consumption. As depicted in Fig. 8b, the continuous dissolution of Ca$_3$SiO$_5$ was observed throughout the hydration induction period. Silane treatment notably reduced the hydration degree of Ca$_3$SiO$_5$, resulting in a higher relative content of Ca$_3$SiO$_5$ compared to the untreated (blank) group at the same time points. TEM images further confirmed the reduced dissolution rate induced by silane treatment in silane-treated Ca$_3$SiO$_5$ paste (Fig. 8c). After 10 minutes of hydration, a discernible precipitation layer was formed and coated on the surface of Ca$_3$SiO$_5$ particles in the blank

group (Layer A). In contrast, silane-treated Ca$_3$SiO$_5$ exhibited a thinner precipitation layer (Layer B), and in some regions, the precipitation layer was entirely absent. Additionally, hydration gradually disrupted the Ca$_3$SiO$_5$ particle lattice, forming disordered, layered C-S-H (calcium silicate hydrate) products on particle surfaces in the blank group within 60 min. However, this transformation was postponed in the silane-treated group, appearing until 120 min, when the nearly complete dissolution of Ca$_3$SiO$_5$ particles was observed in the blank group.

## Discussion

Our research introduces a nano-level discovery that the dissociation and adsorption of silane on the Ca$_3$SiO$_5$ surface markedly hinders calcium ion dissolution. This suggests that a key mechanism responsible for the delayed early hydration of cement caused by SCAs may be their ability to reduce the dissolution rate of clinker phases. This reduction slows the accumulation of calcium concentration in the solution, delaying the precipitation threshold and thereby extending the induction period. Such retardation may be one of the primary contributors of the hydration delay caused by SCAs. It is important to note, however, the mechanisms by which SCAs retard hydration are multifaceted and interconnected. While the inhibition of dissolution proposed here is a contributing factor, other mechanisms, such as the complexation or chelation of SCAs with calcium ions in solution, or the inhibition of nucleation and growth of the hydration product C-S-H, may also play significant roles[16,18,42]. Additionally, the Ca$_3$SiO$_5$ surface model used in this study represents an idealized system, not accounting for surface defects like dislocations. These defective surfaces are crucial to hydration reactions, as they provide numerous active sites for water molecule adsorption[43,44]. At these defects, calcium ions experience a relatively unsaturated coordination environment, resulting in their incomplete dissolution pathways as compared to a perfect surface. While the five-coordination calcium dissolution pathway proposed in this study is also applicable to defect surfaces,

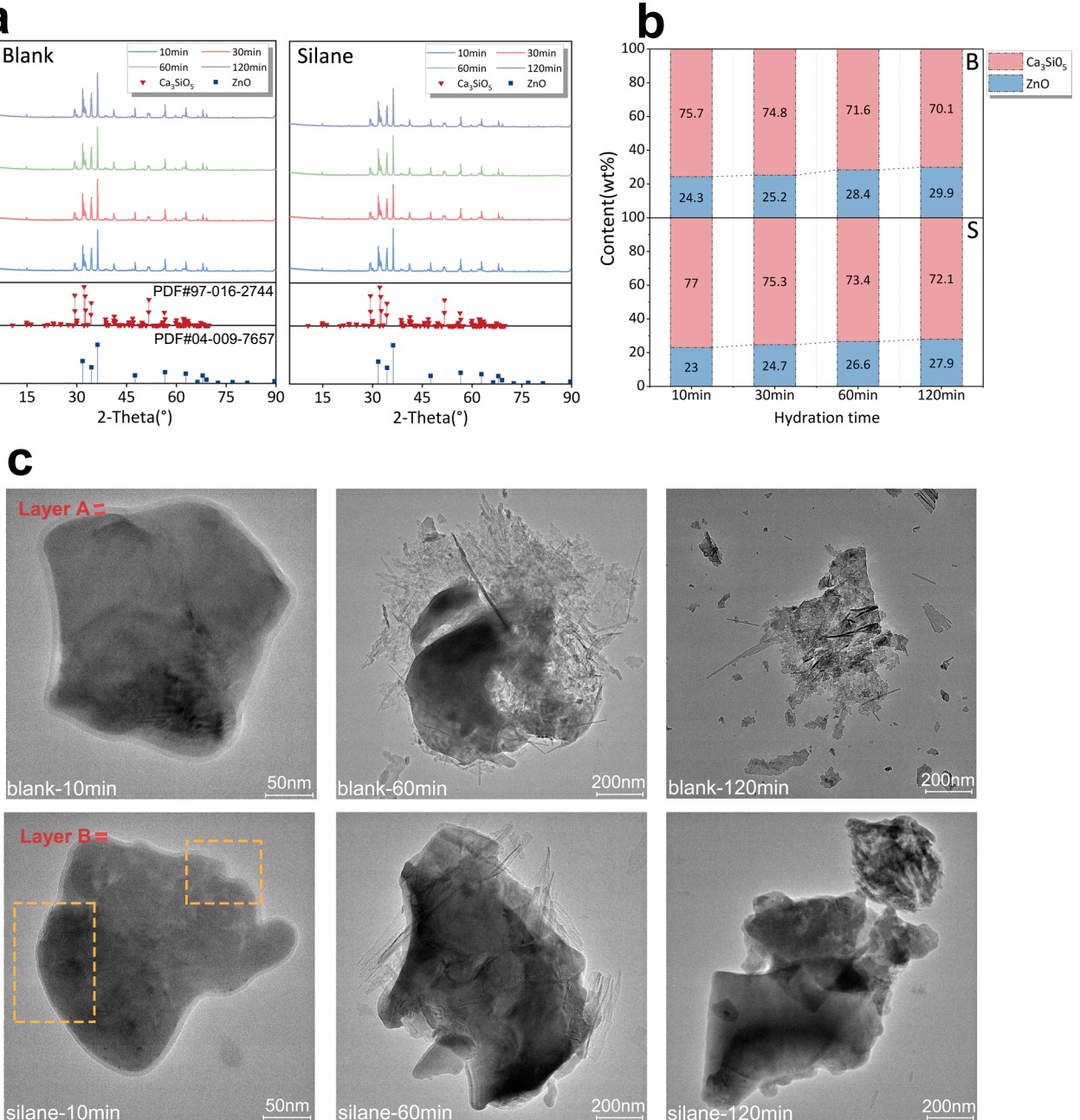

**Fig. 8 | Phase structure analysis during early hydration stages. a** QXRD curves at various hydration intervals, with standard PDF cards for Ca₃SiO₅ and ZnO highlighted. **b** Phase content derived from XRD fitting analysis, where the R-value for each group is maintained below 12%. B denotes the pure water group, and S represents the silane-treatment group, with phase content expressed as the weight percentage (wt%) of the sample. **c** Representative TEM images at three distinct hydration times, where yellow dashed boxes highlight regions of dissolution inhibition on the surface of Ca₃SiO₅ particles in the silane-treatment group. The red lines mark the sedimentation of Layers A and B.

the adsorption behavior of silane molecules at defect sites may be different. For instance, silane might adsorb near such vacancies and prevent pit opening, a hypothesis that warrants further investigation. Furthermore, the hydrolysis and condensation of silane may influence their inhibition effect. Hydrolysis of the silane monomer promotes its adsorption onto the surface of Ca₃SiO₅, as the -SiOH group provides additional sites for the formation of Ca-O and hydrogen bonds. As a result, silane hydrolysis plays a critical role in inhibiting the dissolution of calcium ions from the surface. In terms of condensation, silane oligomer formed through self-condensation may increase steric hindrance, further preventing the dissolution of Ca ions. However, self-condensation also decreases the number of -SiOH groups, weakening

the adsorption strength on the Ca₃SiO₅ surface. In contrast, condensation between the silanol group and Si-OH on the substrate surface leads to the formation of Si-O-Si covalent bonds[22], thereby enhancing surface adhesion and inhibiting the dissolution.

By elucidating the molecular mechanisms underlying silane's interaction with cementitious phases, this study provides valuable insights into the hydration behavior of organic-modified cement systems. Silane's demonstrated ability to stabilize calcium ions and hinder dissolution highlights its potential to enhance the durability, mechanical performance, and chemical resistance of cement-based materials. This work establishes a theoretical foundation for designing next-generation organic-inorganic composites with improved

performance characteristics, paving the way for more sustainable and functional construction materials.

In summary, here we demonstrate that silane effectively inhibits the dissolution of $Ca_3SiO_5$ at the nanoscale, as revealed through a combination of ab initio simulations and experimental validation. We found that silane molecules preferentially adsorb onto calcium sites on the $Ca_3SiO_5$ surface via dissociative adsorption, creating a steric hindrance effect. This occurrence prevents water molecules from interacting with the calcium ions, thereby increasing the resistance to calcium dissolution. The most stable structure of calcium in the silane-treated system is penta-coordinated with surface oxygen atoms, compared to the tri-coordinated structure when exposed to aqueous solutions, indicating that silane significantly stabilizes the surface and inhibits calcium release. Furthermore, the shielding effect of silane alters the free energy changes of stepwise calcium dissolution from negative to positive, resulting in the transformation of the dissolution reaction from spontaneous to non-spontaneous. Experimental results provide further validation of simulation findings. The hydration heat curve demonstrated that silane markedly suppressed the early hydration process of $Ca_3SiO_5$, particularly during the dissolution phase. XPS spectra confirm the presence of chemical interactions between silane molecules and $Ca_3SiO_5$ particles, validating the proposed surface interaction model. ICP-OES analyses reveal a substantial reduction in the calcium ions concentration following silane incorporation. TEM analysis suggests that the silane treatment reduced the thickness of the precipitation layer on the surface of $Ca_3SiO_5$ particles and inhibited the growth of hydration products. Collectively, these results suggest that the dissolution of calcium ions is effectively impeded due to interfacial interactions between silane and the $Ca_3SiO_5$ surface. These findings have broader implications for the development of organic-inorganic cement composites.

## Methods
### Atomistic model construction
In industrial clinker, the primary phase $Ca_3SiO_5$ exists in two monoclinic crystal forms: M1 and M3[45]. In this study, the M3 polymorph (reported by ref. 46) was selected to develop an atomic model. The (111) crystal plane of the M3 unit cell was selected to construct the surface model, as the literature indicated that cross-sections at varying orientations to the (111) plane exhibited lower surface energies than other planes[47–49]. The dimensions of the initial $Ca_3SiO_5$ slab model were set to $14.31 \times 11.96 \times 14.95$ Å along the a, b, and c axes. A 20 Å vacuum layer was introduced above the surface to avoid interactions between periodic images. The system was then subjected to geometric optimization via the DFT method[50]. Next, a water layer composed of 100 $H_2O$ molecules with a density of $1.0\,g/cm^3$ (dimensions were $14.00$ Å $\times 11.40$ Å $\times 22.04$ Å) was added to the $Ca_3SiO_5$ surface. Subsequently, a 2 ps AIMD simulation with 1.0 fs time step was conducted to relax the $Ca_3SiO_5/H_2O$ solid-liquid interface for adjusting the positions of molecules. To ensure thorough contact at the solid-liquid interface[23,24], the surface OH group density following relaxation was calculated and is presented in Supplementary Fig. 2. For the silane-containing system, an APTES molecule after complete hydrolysis was introduced into the water layer, as shown in Supplementary Fig. 1.

### Adsorption tests
Geometric optimization of the (111) $Ca_3SiO_5$ slab was carried out by CP2K code with the Gaussian and plane wave (GPW) method[51]. The exchange-correlation potential, encompassing electronic interactions, is calculated using the generalized gradient approximation (GGA) with the Perdew–Burke–Ernzerhof (PBE) functional[52]. Core electrons were modeled with the Goedecker–Teter–Hutter (GTH) pseudopotential, while valence electrons were described through a mixed GPW approach. The Brillouin zone was sampled using the gamma-point approximation, and wavefunctions were expanded on a double-zeta

valence polarized (DZVP) basis set, complemented by an auxiliary plane wave basis set with a cutoff energy of 450 Ry. Grimme's D3 correction[53] was performed to address dispersion effects. The optimization threshold was 0.02 eV Å$^{-1}$, and the global convergence criterion of energy for the electronic self-consistent loops was set as $10^{-5}$ eV.

Following the optimization of the $Ca_3SiO_5$ substrate, the hydrolyzed APTES monomer was positioned on different reactive sites on the $Ca_3SiO_5$ surface. Considering that SCAs adsorption is primarily governed by the silicon group, surface calcium, and oxygen atoms (in silicate tetrahedra) were selected as the potential adsorption sites[54]. The initial distance was set to 3 Å for the adsorption between surface calcium ions and APTES oxygen atoms, and 2.5 Å for the interaction between the APTES silicon group and the silica tetrahedra on the $Ca_3SiO_5$ surface. The optimization parameters were kept consistent with those for bulk optimization[55].

### AIMD simulations
The lack of a reliable force field for the systems containing C/H/O/N/Ca/Si led to the decision to use AIMD for accurately simulating the adsorption of silane monomer to tricalcium silicate and its effect on dissolution behavior, which offers superior accuracy by capturing chemical reactions and electron transfers. The AIMD simulations in this work were performed using the Quickstep module within the CP2K software[56], utilizing the same DFT framework as the geometric optimization. Throughout the simulations, the nuclei were treated under the Born–Oppenheimer approximation[57]. The simulations were conducted at a temperature of 300 K, controlled via the Nosé-Hoover thermostat with a time constant of 100 fs in the canonical NVT ensemble[58]. The AIMD time step was initially set at 0.5 fs for relaxation, while a time step of 1.0 fs was used for WT-MetaD simulations, final structural analysis, and subsequent MetaD simulations. The final structural analysis simulations are with the replacement of hydrogen by deuterium to accelerate the structural evolution without energy drifts. The total AIMD simulation time was presented in Supplementary Table 3

### Metadynamics simulations
WT-MetaD[59] simulations based on ab initio method were conducted to sample the configurations occurred during dissolution of the surface calcium cation under the influence of APTES. As a comparison, the dissolution pathway of Ca in the pure aqueous solution was also performed.

In WT-MetaD simulations, the two-dimensional collective variables (CVs) were characterized by defining the coordination number of the calcium cation. The first CV, $CN(Ca-O_s)$ corresponds to the coordination number between the monitored calcium cation and the oxygen atoms within the $Ca_3SiO_5$ slab. The second CV, $CN(Ca-O_{w/w+a})$ represents the coordination number between the monitored calcium cation and all other oxygen atoms in the system ($O_w$ or $O_{w+a}$, where $w$ refers to water oxygens and $a$ to APTES oxygens, respectively). The calculation of these coordination numbers (CNs) can be expressed as follows:

$$CN(Ca, O_{w/w+a}) = \sum_{i \in A} \sum_{i \in B} s_{ij}(r_{ij}) \tag{1}$$

Where $s_{ij}$ takes the value of 1 when a contact between atoms $i$ and $j$ established, and 0 otherwise. To maintain the continuity of the calculated CV and ensure smooth derivatives, $s_{ij}$ is typically substituted with a switching function. The standard form of this switching function is as follows:

$$s_{ij}(r_{ij}) = \frac{1 - \left(\frac{r_{ij}-d_0}{r_0}\right)^n}{1 - \left(\frac{r_{ij}-d_0}{r_0}\right)^m} \tag{2}$$

Here, $r_{ij}$ represents the distance between atoms $i$ and $j$, while $d_0$ denotes the central value of the function. The parameter $r_0$ defines the acceptance distance for the switching function, where the function value becomes $n/m$ at $d_0 + r_0$. In this work, $d_0$ represents the distance between monitored calcium and oxygen atoms, defined as 2.42 Å based on the equilibrium Ca–O bond length as reported in the literature[60,61]. The value of $r_0$ is set to 0.4 Å, approximately half of the full width at half maximum of the radial distribution function for Ca–O[62,63]. The parameters $n$ and $m$ are assigned values of 6 and 12, respectively.

WT-MetaD simulations were conducted with the help of the PLUMED software[64]. In both simulations with and without SCAs molecule, Gaussian hills with 0.15 width and 3.5 kJ·mol$^{-1}$ initial height were periodically added along two CVs every 30 timesteps, and the bias-factor was set to be 15. Constraints were imposed on CN(Ca-O$_{w/w+a}$) in the range [0.5, 6], and CN(Ca$_s$-O$_s$) in the range [0.5, 4.5]. This is achieved by introducing boundary walls with a quadratic form where the coefficient is 500 kJ/mol.

To evaluate the convergence of WT-MetaD simulations, we analyzed the time-dependent estimates of the free energy. In principle, upon reaching convergence, the reconstructed free energy surfaces (FES) should exhibit similar profiles, differing only by a constant offset. Accordingly, we computed the FES estimate for every 100 deposited Gaussian kernels (3 ps). The FES profiles at different time intervals should remain consistent, aside from the constant offset. Detailed convergence information is provided in Supplementary Fig. 3 and Fig. 4 (Supplementary Note 3). Convergence of the WT-MetaD simulations were achieved after sampling 66,000 configurations in the system containing APTES and 63,000 configurations in the blank group, respectively.

### Raw materials

Tricalcium silicate, with a purity of 98.08%, was acquired from Wuhan Kabuda Chemical Co., Ltd. Its chemical composition is determined by X-ray fluorescence analysis, as listed in Supplementary Table 2 (Supplementary Note 4). 3-aminopropyl triethoxysilane with a purity of 99% was obtained from Shanghai Macklin Biochemical Co., Ltd.

### Hydration experiments

APTES was firstly pre-hydrolyzed to form a silanol group before being added to the $Ca_3SiO_5$ paste The APTES solution was prepared by adding APTES into the deionized water (DI), where liquid to solid ratio (l/p) of 5.0 was used and 1% APTES by weight of $Ca_3SiO_5$ was used. Then the solution was stirred by a magnetic stirrer at 500 rpm for 1 hour to accelerate the hydrolysis. In the silane treatment $Ca_3SiO_5$ paste, $Ca_3SiO_5$ powder was weighed and subsequently mixed with the APTES solution while the l/p ratio remained unchanged with only water in the blank group. The resulting paste underwent rapid magnetic stirring at 100 rpm for 40 s to ensure uniform mixing of the solution and the powder, followed by curing of the paste. At specific hydration intervals of 10, 20, 40, 50, 60, 75, 90, and 120 min, samples of the $Ca_3SiO_5$ paste were collected and filtered by suction using a 45-micron filter to separate the solid and liquid phases. For ICP-OES measurements, the resulting clarified hydrolysate was further diluted by a factor of 20 using DI, followed by the addition of HNO$_3$ (6.3% by volume) for acidification. For XPS measurements, samples were hydrated for 10, 30, 60, and 120 min. Afterward, the solid and liquid phases were separated by suction filtration using a 45-micron filter. The solid phase was collected and immediately flash-frozen with liquid nitrogen to halt hydration, then freeze-dried under vacuum to eliminate residual moisture. The resulting dry powder was ground and prepared for XPS measurements. For the QXRD test, the tested powder was prepared by blending the freeze-dried solid powder with analytical reagent-grade zinc oxide (ZnO, 99.99%, Aladdin) at a mass ratio of 4:1. For TEM, a small amount of solid powder was mixed with anhydrous ethanol and

subsequently dispersed using ultrasonication. The suspended liquid droplets were then deposited onto a carbon film carrier membrane and dried in a vacuum dryer to prepare the TEM samples.

### Hydration kinetics characterization

The concentration of the dissolved calcium ions was analyzed by ICP-OES (Agilent 720 ES ICP-OES System, USA). After the sample was introduced into the instrument, it was analyzed under an RF power of 1.2 kW. The plasma flow, auxiliary flow, and nebulizer flow rates were set at 15.0, 1.50, and 0.75 L/min, respectively. Both the sample uptake delay and instrument stabilization delay were configured to 15 s.

XPS was performed to investigate the evolution of the chemical environments of Ca and Si in silane-treated $C_3S$ paste during hydration. XPS analysis was conducted using a Thermo Scientific K-Alpha instrument equipped with an Al Kα X-ray source (hv = 1486.6 eV), with a power of 72 W (12 kV, 6 mA). The analysis was performed on a 400 μm diameter area under high vacuum conditions (<$2.0 \times 10^{-7}$ Pa). An appropriate amount of the sample was pressed onto the sample tray before being introduced into the XPS system. For this study, the pass energy for the full spectrum scan was set to 150 eV with a step size of 1 eV, while the narrow-spectrum scan was performed with a pass energy of 50 eV and a step size of 0.1 eV. The binding energy of the C 1s peak at 284.8 eV, corresponding to adventitious carbon[65], was utilized to correct for charge effects in the spectrum, followed by deconvolution analysis.

All QXRD tests in this study were performed using a Rigaku SmartLab 9 kW upgraded apparatus with CuKα radiation (λ = 1.5418 Å) operated at 45 kV and 200 mA. The powder samples were scanned over a $2\theta$ range of 5.00° to 90.00° with a step size of 0.02° and a scanning rate of 2°/min. The resulting spectra were refined and quantitatively analyzed using Jade 9.0 software.

The kinetics of $Ca_3SiO_5$ hydration were monitored using an eight-channel isothermal calorimeter (I-Cal 8000 HPC, Calmetrix). $Ca_3SiO_5$ pastes with a w/p ratio of 0.5 were prepared, both with and without 1% wt APTES to the $Ca_3SiO_5$ powder. Heat evolution was monitored for the first 3 days to track the hydration process.

The morphology of $Ca_3SiO_5$ after 10-, 60-, and 120-minute hydration intervals was examined using a TEM (FEI TALOS F220S). TEM images were analyzed using DigitalMicrograph software.

### Inclusion and ethics

This study did not involve any human participants or personal data. All data generated and analyzed during this study are included in this published article and its supplementary information files.

## Data availability

All relevant data were available in the main text and Supporting Information and can be obtained from the authors upon request. Source data are provided with this paper.

## Code availability

All the software used in this work is open source. No specific software was developed for this work.

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

## Acknowledgements
The authors gratefully acknowledge the financial support provided by Science and Technology Development Fund (FDCT), Macau SAR (0067/2022/A2) [Chen], Research & Development Office at University of Macau (SRG2022-00046-IAPME and MYRG-GRG2023-00220-IAPME-UMDF) [Chen], Macao Science and Technology Development Fund (FDCT) (Grant No. 0074/2023/RIB3) [Li], the Macau University of Science and Technology Faculty Research Grant (Grant No. FRG-24-085-FIE) [Li]. The author would like to thank Beijing PARATERA Tech CO., Ltd. and Bohrium Cloud Platform of DP Technology for providing resources for simulations.

## Author contributions
B.C. and Y.L. conceived and supervised the project. M.W., B.C., and Y.L. designed the research. M.W. performed the simulations, analyzed the simulation results, and wrote the manuscript. B.C., M.W., Y.L., H. M., and Y. Z. discussed the results. Y.L., M.W., and B.C. edited the manuscript before submission.

## Competing interests
The authors declare no competing interests.
