## [Transparent Peer Review file · Nature Communications]

Molecular Elucidation of Cement Hydration Inhibition by Silane Coupling Agents

Corresponding Author: Professor Yunjian Li

Version 0:

Reviewer comments:

Reviewer #1

(Remarks to the Author)

This manuscript investigates how SCAs (silane coupling agents) can inhibit the dissolution of tricalcium silicate (C_3S , Ca_3SiO_4), which reacts with water to form cement paste. The study combines simulations at the atomic scale and a bit of experiments. The results showed that silane molecules adsorb onto calcium sites on the Ca_3SiO_5 surface, creating a steric hindrance effect that prevents water molecules from interacting with the calcium ions, thereby increasing the resistance to calcium dissolution. Experimental results confirmed that silane significantly slowed down the early hydration process of Ca_3SiO_5 , particularly during the dissolution phase.

Overall the manuscript is interesting and well written. Despite the fact that SCAs are known to inhibit hydration, the authors propose an interesting approach to try to determine the mechanisms behind it. The methods that are now becoming quite common in atomistic simulation studies but they are used in a seemingly proper way.

I recommend major revisions before considering the manuscript for publication in Nature Communications. In terms of target audience, I think the manuscript is mainly focused on the cement community.

I have some comments and suggestions for the authors:

- It is discussed in the manuscript but a limitation is the simplicity of the model that is considered for the mineral and the solution that is pure water. The choice of the model is probably motivated by the choice to do the simulations using ab initio calculations and I wonder/regret that the authors have not decided to use a classical force field instead, at least for the dissolution simulations. This should be discussed in the manuscript in more details.
- In the abstract, I'd recommend to give some examples of "atomic mechanisms" that the authors are referencing to in the first sentence.
- I am not very convinced by a RMSD calculation performed on 10ps simulations, it's extremely short. I also don't think this figure / calculation provides additional and strong information.
- On the RDF, why is there a peak at 1 Angstrom for the O-H RDF if the O are the atoms from the slab? It looks like the atoms from water molecules have been considered as well. Also, does the two RDF (for blank / silane) converge to the same value at long distance? I would understand that it doesn't converge to 1 because it's at the interface but it should converge to the same value. This should be revised and the conclusions checked again.
- The authors should explain better the choice of the collective variables, in particular the second one which is the coordination number with all O atoms of the system. In my point of view, calcium should have this coordination number quite constant (close to 6) during all the dissolution process because O from the slab will be replaced by water molecules. I do not understand why this CV is varying so much during the metadynamics according to the supplementary materials.
- Does the Ca gets completely solvated in the simulations? What is the evolution of the CVs over time?
- On the S.I. it is showing the convergence of the WTMTD but the authors should also provide the evolution of the CVs to

show that at the end of the simulation the CVs can be fully explored without adding more bias, which I doubt. It looks like the FES is converging only because of the well tempering. This could strongly affect the results.

- The manuscript does a good job of summarizing the findings and their implications, it could benefit from a more thorough discussion of potential future research directions. For example, the manuscript notes that the dissolution of calcium ions is effectively impeded due to interfacial interactions between silane and the Ca_3SiO_5 surface. It would be interesting to explore how these interfacial interactions evolve over time during the hydration process, and how they affect the overall hydration kinetics of the cement.

Minor changes:

- $14.09 \times 11.60 \times 14.95 \text{ \AA}$ should be $14.09 \times 11.60 \times 14.95 \text{ \AA}^3$

- Notations need more consistency in the format that is chosen. For example choose if there is a space between numbers and units: 20 \AA or $20 \text{ \AA}^?$

Reviewer #2

(Remarks to the Author)

The manuscript NCOMMS-24-64340 « Molecular Elucidation of Cement Hydration Inhibition by Silane Coupling Agents » presents an investigation on the dissolution of tricalcium silicate (C3S) in the presence of silane coupling agents (SCAs). Well-tempered ab initio metadynamics simulations were performed to simulate the reaction pathways, free energy changes and free energy barriers associated to the dissolution of calcium ions at the (111) surface of M3 C3S with and without the presence of 3-aminopropyl triethoxysilane (APTES) on the surface. Furthermore, the authors assessed experimentally the dissolution of C3S in the presence of APTES. The results of this study are original and of importance for the fundamental understanding of C3S dissolution in the presence of SCAs. The manuscript is well written and very interesting. Yet, I found few items that should be tackled in order to improve the quality of the manuscript:

1) l. 101-103: "The (111) surface was cleaved from the bulk structure to construct the slab model based on existing literature indicating that the (111) surface of Ca_3SiO_5 possesses greater surface stability and higher chemical reactivity with water compared to other surfaces."

All references except Noirfontaine et al. 2006 are relevant because they report results on surface energy calculated from DFT or MD or static molecular mechanics. However, only Durgun et. al 2014 stated that the (111) is the more stable. Here the authors state "greater" without comparing with other surfaces. Saying that the (111) is rather stable compared to other surface is quite reasonable. However I think that the relation between surface stability and chemical reactivity is not correct. Generally, unstable surfaces with high surface energy have higher tendency for adsorption and are more likely to react with chemical species in order to reach their lower energy.

2) l.162-165: "The r_0 was set 0.4 \AA , which is the difference between the truncation radius of the Ca-O bond in radial distribution function."

I think that more details should be provided on how the r_0 parameter of the switching function was chosen. Did the authors mean that it is the difference between the truncation radius and the position of the first peak, or between the first peak and the next minimum? I am not sure whether the references provide explanations on the chosen values.

3) I think it is important to remember what is defined as Ca_α and Ca_β in terms of structure and coordination. In previous work from the authors, Ca_α is defined as three-coordinated so I did not understand the initial state is not (X,3) in the simulations? It is the case in ref. 27. Additionally, it should be interesting to explain the difference between the simulations of ref. 27 and the simulation in this study with water only. The reaction pathway seems different but the mechanisms are similar?

4) Some few other questions and details:

- l. 133-134 The initial distance was set to be 3 \AA for physical adsorption and 2.5 \AA for chemical adsorption. If possible, can you explain in few words why these distances and why they are different for physical and chemical adsorption.

- I think that the acronym and DS is used without definition to qualify the dissociative adsorption and should be replaced by DA (see p.13 and check the whole manuscript). The same is true for MS and MA (molecular adsorption).

- l. 243-245: "Notably, the proton transfer that occurs in the DS process implies the occurrence of chemisorption, with more adsorption sites occupied and could impede the adsorption of water molecules nearby.". Add spaces ("and could" + "molecules nearby").

- l. 292-293 "In the DA condition, the O_a atom forms double Ca- O_a bonds with a -ICOHP value of 0.46, compared to 0.41 in the MA, indicating a stronger interaction.". The 0.41 value is not corresponding to what is shown in Fig. 3d.

- l.377-378 "The lateral adsorption configuration of APTES molecules occupies multiple surface sites, thereby impeding the diffusion of water molecules to the calcium ion.". This sentence is not directly related to the previous one and the reader may be a bit lost. Maybe add "Furthermore,"?

Reviewer #3

(Remarks to the Author)

This is a comprehensive and well-structured study that combines computational simulations with experimental validation to investigate the inhibitory effects of silane coupling agents (specifically 3-aminopropyl triethoxysilane, APTES) on the early hydration of tricalcium silicate (Ca_3SiO_5). However, the molecular dynamics of cement hydration and the retarding mechanisms of silane coupling agents have been extensively studied in previous literature. While this work presents an interesting approach, it does not sufficiently build upon or differentiate itself from existing knowledge in the field. In particular, the hydrolysis and condensation mechanisms of polysiloxane, which are crucial to understanding the full picture of silane effects on cement hydration, have not been adequately addressed. Some questions and suggestions are as following:

#1: In the computational setup, why was the (111) surface of Ca_3SiO_5 chosen for the slab model? Are there other surfaces that might be relevant for this study?

#2: The study focuses on APTES as the silane coupling agent. How might the results differ for other types of silanes commonly used in cement applications?

#3: In the experimental section, why was a 1% concentration of APTES chosen? How might varying concentrations affect the results?

#4: The paper mentions that silane treatment may inhibit the nucleation and growth of C-S-H. Could the authors elaborate on the potential long-term effects of this inhibition on cement properties?

#5: The authors have not sufficiently contextualized their work within the existing body of literature on silane effects in cement hydration. For instance, the statement on lines 58-60 that "the inhibition mechanisms of SCAs on cement hydration remain elusive" overlooks significant prior work in this area. A more comprehensive review of existing studies on silane retardation mechanisms is necessary.

#6: While the ab initio metadynamics simulations provide interesting insights, it's not clear how these findings significantly advance our understanding beyond existing knowledge. The authors should clearly articulate how their results on calcium dissolution pathways (as shown in Figures 4 and 5) differ from or build upon previous studies.

#6: The experimental methods section would benefit from more detail on sample preparation, particularly the mixing procedure for the Ca_3SiO_5 pastes with and without APTES.

#7: The paper lacks a comprehensive mechanistic model that integrates both the observed dissolution inhibition and the known hydrolysis/condensation reactions of silanes. This is necessary to provide a full picture of silane effects in cement systems.

#8: Consider adding a schematic diagram illustrating the proposed mechanism of silane inhibition on Ca_3SiO_5 dissolution. This would help readers visualize the key findings.

#9: The conclusion could be strengthened by explicitly stating the broader implications of this research for the development of organic-inorganic cement composites.

#10: For the ICP-OES analysis (Fig. 9), provide more data points, especially in the early stages of hydration (0-20 minutes) to better capture the initial dissolution kinetics.

#11: In the XPS analysis (Fig. 10), include quantitative data on peak shifts and intensities. A table summarizing these changes across different hydration times would be valuable.

#12: For the heat flow measurements (Fig. 8), extend the monitoring period beyond 3 days to capture any long-term effects of silane on hydration kinetics.

#13: Include Rietveld analysis of XRD data to quantify the phase composition changes during hydration, which would complement the XPS and calorimetry results.

#14: Provide more details on the AIMD simulations, including total simulation time and time step used. This information is crucial for assessing the reliability of the computational results.

#15: Include additional characterization techniques such as High resolution TEM to visualize the microstructural changes induced by silane treatment, particularly at the Ca_3SiO_5 -silane interface, to provide a quantitative comparison between their observed retardation effects and those reported in previous literature.

#16: Consider performing isothermal titration calorimetry (ITC) experiments to directly measure the binding energies between silane and Ca_3SiO_5 surfaces, which could validate the computational adsorption energy calculations.

Version 1:

Reviewer comments:

Reviewer #1

(Remarks to the Author)

The authors have answered all my comments carefully, I think the methodology is sound and the results are interesting. My only concern is about the appeal for a broad audience because I think this is very targeted to wards the cement community, but, otherwise, I recommend the paper for publication.

Reviewer #2

(Remarks to the Author)

The authors addressed all my comments with care and I think that the manuscript is now ready to be published. Here are corrections of typos:

1) plane exhibited lower surface energies than other planes (l.113-114)

2) compared to 0.393 in MA process (l. 317)

Reviewer #3

(Remarks to the Author)

I have reviewed the authors' responses to my previous comments and concerns regarding manuscript NCOMMS-24-64340A. The authors have thoroughly addressed all points raised in my review through detailed responses and appropriate revisions to the manuscript. The authors have made substantive improvements to both the technical content and clarity of the manuscript. The revised version presents a more complete and rigorous analysis of silane coupling agents' effects on cement hydration. Given these thorough revisions and the scientific merit of their work, I recommend this manuscript for publication in Nature Communications.

Dear Editors and Reviewers,

Thank you very much for your hard work and vacomments on our manuscript “Molecular Elucidation of Cement Hydration Inhibition by Silane Coupling Agents” (Manuscript ID: NCOMMS-24-64340A). After studying the comments carefully, we have made the modifications. The *Revised Manuscript* and the *Revised Manuscript with Track Changes* have been submitted. Here are our **point-to-point responses** to the reviewers’ comments.

Reviewer #1

Comment 1: It is discussed in the manuscript but a limitation is the simplicity of the model that is considered for the mineral and the solution that is pure water. The choice of the model is probably motivated by the choice to do the simulations using ab initio calculations and I wonder/regret that the authors have not decided to use a classical force field instead, at least for the dissolution simulations. This should be discussed in the manuscript in more details.

Response: Many thanks for your comment. We fully understand your concern about the choice of ab initio methods instead of the classical force field molecular dynamics (CMD) simulations. The difference between the ab initio molecular dynamics (AIMD) used in this work and CMD is how to calculate the potential energy of the system and forces between atoms. Obviously, the accuracy of the simulation result is higher for AIMD than CMD. Additionally, the interactions between the silane monomer and the tricalcium silicate surface, as well as the dissolution processes, involve complex chemical reactions, including bond formation and breaking, proton transfers, and potentially the hydrolysis of silane. These chemical reactions involves electron transfer between atoms, which AIMD can accurately capture through first-principles calculations. In contrast, CMD relies on predefined force fields that lack the flexibility to model bond rearrangements or accurately describe complex chemical reactions. Although reactive force fields like ReaxFF have been widely used to study some chemical reactions. However, its effective application requires careful consideration of the availability of parameter sets and training data for parameterization. In fact, ReaxFF relies on parameter sets tailored to specific combinations of elements and their interactions. In cement system, the commonly used ReaxFF is developed by Hegoi Manzano and coworkers, which is only suitable for the Si/Ca/O/H systems (Manzano et al. Langmuir 2012, 28 (9), 4187-4197). To the best of our knowledge, there is currently no ReaxFF force field that can directly and reliably model a system containing all the necessary elements in our system containing C/H/O/N/Ca/Si. That is why although there are numerous studies on reactive MD simulations on C-S-H/tri(di)calcium silicate system^{1 2}(Xu et al. Nature Communications, 2024, 15(1): 2731 and Zheng et al. Applied Surface Science, 2025, 680: 161443), there is currently no study on reactive MD simulation on the organic molecules/C-S-H system.

Therefore, we opted for an *ab initio* approach to accurately simulate the the effect of the adsorption of silane monomer to the surface of tricalcium silicate on its dissolution behavior.

The lack of a reliable force field for systems containing C/H/O/N/Ca/Si led to the decision to use AIMD for accurately simulating the adsorption of silane monomer to tricalcium silicate and its effect on dissolution behavior, which offers superior accuracy by capturing complex chemical reactions and electron transfers

(Line 148, Page 8)

Comment 2: In the abstract, I'd recommend to give some examples of "atomic mechanisms" that the authors are referencing to in the first sentence.

Response: Thank you for the insightful advice, we fully agree with that adding some examples will improve the readability of the paper. As a result, we have added the corresponding examples of **the adsorption of silane molecules onto reactive surface sites and modification of ion detachment pathways** in this part, as you can see in Line 15:

Silane coupling agents (SCAs) are known to retard early hydration when incorporated into fresh cement paste, yet the atomic-level mechanisms, such as the adsorption of silane molecules onto reactive surface sites and modification of ion detachment pathways, underlying their influence on clinker dissolution remain largely unexplored.

(Line 14, Page 2)

Comment 3: I am not very convinced by a RMSD calculation performed on 10ps simulations, it's extremely short. I also don't think this figure/calculation provides additional and strong information.

Response: Thank you for your feedback on the RMSD calculation. I understand your concerns regarding the short 10ps simulation. Our original intention is to use this to judge the balance of MD. According to your recommendation, we removed this figure in the manuscript.

Comment 4: On the RDF, why is there a peak at 1 Angstrom for the O-H RDF if the O are the atoms from the slab? It looks like the atoms from water molecules have been considered as well. Also, does the two RDF (for blank / silane) converge to the same value at long distance? I would understand that it doesn't converge to 1 because it's at the interface but it should converge to the same value. This should be revised and the conclusions checked again.

Response: Thank you for your valuable comment. The peak at 1 Å in the O-H RDF arises from proton exchange between water molecules and the silicate tetrahedron ($-\text{SiO}_4$), leading to hydroxylation of

the Ca_3SiO_5 surface. Regarding RDF convergence, the curves reach similar values within the same system, as illustrated in the figure below, where (a) represents the silane system and (b) represents the blank system. In the manuscript, these RDFs are presented separately to highlight the differences between the two systems.

Besides, the RDF curves have been also checked and revised in the Manuscript:

(b) the radial distribution function (RDF) between calcium ions and water oxygen atoms; and (c) the RDF between slab oxygen atoms and water hydrogen atoms. Dotted lines in panels (b) and (b) represent the integration curves of the RDFs. The ordinate values at the intersection points between first RDF valley and integrals indicate the average coordination numbers for the respective pairs in the system.

(Line 423, Page 24)

Comment 5: The authors should explain better the choice of the collective variables, in particular the second one which is the coordination number with all O atoms of the system. In my point of view, calcium should have this coordination number quite constant (close to 6) during all the dissolution process because O from the slab will be replaced by water molecules. I do not understand why this CV is varying so much during the metadynamics according to the supplementary materials.

Response: Thank you for your comment. Regarding to your comment ‘the second one which is the coordination number with all O atoms of the system’, we have to clarify that the the second CV is

defined as the CN between calcium ions and the total oxygen atoms in APTES and water molecules, represented as $CN(\text{Ca-O}_{w+a})$, while the first collective variable (CV) is defined as the coordination number (CN) between calcium ions and oxygen atoms within the Ca_3SiO_5 slab, denoted as $CN(\text{Ca-Os})$. These CVs were selected based on the observation that both APTES and water molecules can dissociatively adsorb and form Ca–O bonds on the Ca_3SiO_5 surface. We fully agree with your opinion that “calcium should maintain a relatively constant coordination number (close to 6) throughout the dissolution process, as oxygen from the slab is replaced by water molecules.” This perspective aligns with the free energy surface (FES) analysis that the most stable dissolution pathway for calcium ions involves maintaining coordination states of six or seven (sum of the CV1 and CV2). The variations of individual CVs are normal as the system explores different chemical space. While the sum of the two CVs is almost stable around 6 to 7, indicating the coordination environment of the Ca ion during dissolution is unchanged.

Comment 6: Does the Ca gets completely solvated in the simulations? What is the evolution of the CVs over time?

Response: Thank you for your comment. In the current WT-MetaD simulation results, the lowest positions of $CN(\text{Ca-Os})$ in both scenarios are two. This is due to the ever-lowering Gaussian peaks in the well-tempered MetaD simulations limit the exploration of a broader chemical space while guarantee the convergence and accuracy of the free energy surface. Although the current simulations do not achieve complete dissolution of Ca ions from the surface into the solution, the inhibitory effect of silane on Ca ion dissolution is evident. When silane is adsorbed at the reactive site, the stable states shift from (3,3) to (5,1). Furthermore, the dissolution process transitions from spontaneous in a water environment to non-spontaneous in the presence of silane, providing additional confirmation of silane’s inhibitory effect on Ca dissolution.

Inspired by the reviewer’s recommendation, we further performed conventional MetaD simulations, that is without well-tempered settings, to probe the dissolution process at the next stage. In these simulations, all parameter settings were kept consistent with those used in the WT-MetaD simulations, except for the removal of the bias factor. Each system, with and without silane, was simulated for 27 ps under these conditions.

The results are as follows:

a b Two-dimensional free energy surface (FES) landscapes of the silane and blank systems obtained from 27 ps conventional MetaD simulations. **c** The completely dissolved structure observed in the blank system.

For the blank group, conventional MetaD simulations can fully explore the configuration in which calcium ions are completely dissolved. In contrast, for the silane group, the minimum coordination with the surface remains at two. However, we need to point that the free energy surface obtained through standard MetaD exploration cannot converge, rendering the free energy barrier and free energy change between minimum states cannot be quantitatively analyzed. In contrast, the free energy surface obtained using the Well-Tempered method provides a clearer explanation: the Ca ion is more difficult to dissolve in the silane system than in the water system, indicating the silane molecule prevents Ca from leaching surface. Therefore, whether the dissolution of the calcium ion system is complete does not affect the present results.

The time-evolution of CVs during WT-MetaD simulations are as follows:

a b The time evolution of CVs along with the sampling time. $CN(Ca-O_s)$, $CN(Ca-O_w)$, and $CN(Ca-O_a)$ represent the coordination numbers between calcium (Ca) and surface oxygen (O_s), water (O_w), and

APTES oxygen (O_a) atoms, respectively.

Comment 7: On the S.I. it is showing the convergence of the WTMTD but the authors should also provide the evolution of the CVs to show that at the end of the simulation the CVs can be fully explored without adding more bias, which I doubt. It looks like the FES is converging only because of the well tempering. This could strongly affects the results.

Response: Thank you for your comment. The time evolution of the collective variables (CVs) has been added to the revised manuscript to confirm the equilibrium of the WT-MetaD simulations. Furthermore, the simulation durations have been extended: for the APTES system, the simulation time was increased **from 42 ps to 66 ps**, and for the blank system, it was extended **from 27 ps to 63 ps**. The production times for these simulations are detailed in **Table 1**.

Table 1 Production time of AIMD simulations

Production details	Time
Surface relaxation after explicit solution introduced	10 ps
Dissolution pathway of Ca under APTES influence	66 ps
Dissolution pathway of Ca in explicit aqueous solution	63 ps
Configuration analysis of final state relaxation from WT-MetaD	10 ps for each system

As for the CV space, the ever-lowering Gaussian peaks in the well-tempered MetaD simulations limit the exploration of the CV1 from five to two. While the conventional MetaD can explore a broader CV1, the free energy surface cannot converge as we explained in the last comment. Although the current simulations do not achieve complete dissolution of Ca ions from the surface into the solution, the inhibitory effect of silane on Ca ion dissolution is evident. When silane is adsorbed at the reactive site, the stable states shift from (3,3) to (5,1), indicating the Ca ion tends to remain surface state when the silane adsorbs on the Ca site. Furthermore, the dissolution process transitions from spontaneous in a water environment to non-spontaneous in the presence of silane, providing additional confirmation of silane's inhibitory effect on Ca dissolution. The conventional MetaD simulations with the exploration of a full dissolution process confirmed the inhibitory effect of the silane on the Ca dissolution process.

Minor changes:

(1) $14.09 \times 11.60 \times 14.95 \text{ \AA}$ should be $14.09 \times 11.60 \times 14.95 \text{ \AA}^3$.

(2) Notations needs more consistency in the format that is chosen. For example choose if there is a

space between numbers and units: 20Å or 20 Å?

Response: Thank you for your careful comments, we now have revised these unit expressions to make the manuscript clearer.

(1) Next, a water layer composed of 100 H₂O molecules with a density of 1.0 g/cm³ (dimensions were 14.00 Å × 11.40 Å × 22.04 Å) was added on the Ca₃SiO₅ surface.

(Line 118, Page 7)

(2) A 20 Å vacuum layer was introduced above the surface to avoid interactions between periodic images. The system was then subjected to geometric optimization via the DFT method.

(Line 115, Page 6)

Reviewer #2

Comment 1: l. 101-103: “The (111) surface was cleaved from the bulk structure to construct the slab model based on existing literature indicating that the (111) surface of Ca_3SiO_5 possesses greater surface stability and higher chemical reactivity with water compared to other surfaces.”

All references except Noirfontaine et al. 2006 are relevant because they report results on surface energy calculated from DFT or MD or static molecular mechanics. However, only Durgun et. al 2014 stated that the (111) is the more stable. Here the authors states “greater” without comparing with other surfaces. Saying that the (111) is rather stable compared to other surface is quite reasonable. However I think that the relation between surface stability and chemical reactivity is not correct. Generally, unstable surfaces with high surface energy have higher tendency for adsorption and are more likely to react with chemical species in order their lower energy.

Response: Thank you for your insightful comment. We have removed irrelevant references (e.g., Noirfontaine et al., 2006) and revised the for clarity and precision because the low energies here are too subjective. The updated content now reads: “The (111) crystal plane of the M3 unit cell was selected to construct the surface model, as the literature indicates that cross-sections at varying orientations to the (111) plane exhibit low surface energies than other planes.” Additionally, we fully agree that a surface with lower energy may not necessarily exhibit high reactivity with water molecules. Generally, lower surface energy is associated with greater stability, making such crystal faces more likely to form in nature. Consequently, it would be an oversimplification to assume a direct correlation between the (111) surface and its chemical reactivity with water molecules. We have removed this view from the manuscript.

The revised content could be found in the revised Manuscript:

The (111) crystal plane of the M3 unit cell was selected to construct the surface model, as the literature indicates that cross-sections at varying orientations to the (111) Ca_3SiO_5 surface exhibit low surface energies.

(Line 112, Page 6)

Comment 2: l.162-165: “ The r_0 was set 0.4 Å, which is the difference between the truncation radius of the Ca-O bond in radial distribution function.” I think that more details should be provided on how the r_0 parameter of the switching function was chosen. Did the authors mean that it is the difference between the truncation radius and the position of the first peak, or between the first peak and the next minimum? I am not sure whether the references provide explanations on the chosen values.

Response: Thanks for your comment. In this study, the d_0 is the central value of the function, which refers the the central Ca-O bond distance, and the r_0 is the acceptance distance of the switching function, where the function well be n/m at $d_0 + r_0$. Here, we define d_0 is 2.42 Å, **which is the equilibrium bond length between the Ca and O ions³**; r_0 is 0.4 Å, **which is around half of the full**

width at half maximum of the radial distribution function of Ca–O and n and m are 6 and 12⁴, respectively.

The revised content could be found in the revised Manuscript:

The parameter r_0 defines the acceptance distance for the switching function, where the function value becomes n/m at $d_0 + r_0$. In this work, the d_0 represents the distance between monitored calcium and oxygen atoms, defined as 2.42 Å based on the equilibrium Ca–O bond length as reported in the literature. The value of r_0 is set to 0.4 Å, approximately half of the full width at half maximum of the radial distribution function for Ca–O. The parameters nnn and mmm were assigned values of 6 and 12, respectively.

(Line 178, Page 10)

Comment 3: I think it is important to remember what is defined as Ca_α and Ca_β in terms of structure and coordination. In previous work from the authors, Ca_α is defined as three-coordinated so I did not understand the initial state is not (X,3) in the simulations? It is the case in ref. 27. Additionally, it should be interesting to explain the difference between the simulations of ref. 27 and the simulation in this study with water only. The reaction pathway seems different but the mechanisms are similar?

Response: Thanks for your comment. In fact, this study **only considered the dissolution process of pentacoordination calcium ions** from the surface of Ca_3SiO_5 , because compared with low coordination, **the dissolution path of high coordination calcium ions is more complete**, and the influence of silane molecules in the dissolution process can be considered more comprehensively. **The initial state of calcium ions in both systems is (5,1)**. For the blank group, the initial coordination may change due to the influence of surface optimization. Therefore, we reconstructed the surface model for optimization to ensure that the initial dissolved state of calcium ions in both systems is pentacoordinated. Additionally, we increased the sampling time of the well tempered MetaD to ensure that all chemical spaces were explored, as shown in **Fig. 4** and **5**. Furthermore, we also re-analyzed the FES landscape and re-calrified the inhibition mechanisms of APTES on Ca dissolution. The revised content could be found in the revised Manuscript:

The negative adsorption energy of APTES on the Ca_3SiO_5 surface suggests that APTES preferentially adsorbs at reactive sites, potentially influencing the dissolution of Ca_3SiO_5 . To investigate these atomic-level interactions, *ab initio* WT-MetaD simulations were performed to examine the impact of APTES on the dissolution pathway of calcium ions. The two-dimensional free energy surface (FES) depicted in **Fig. 4b** illustrates the free energy barriers along the collective variables (CVs), with the corresponding numerical values represented by the color gradient. The free energy landscape for calcium ions reveals a range of minima distributed over a checkered pattern spanning 120 kJ/mol. These minima are associated with a reduction in the number of bonds to surface oxygen atoms of Ca_3SiO_5 and an increase in bonds with oxygen atoms from APTES. As the simulations progress, the free energy surface reveals an addition/elimination mechanism akin to the hydrolysis process observed in the aluminum phase of Al_2O_3 ⁵. Notably, even when the coordination number of

water molecules and calcium ions is included as a set variable, the coordination bond between the oxygen atoms (O_w) of water molecules and calcium ions is not observed during the sampling process with the addition of Gaussian peaks (**Fig. 4a**).

This absence is attributed to both the hydrophobic effect of APTES and the silicon group adsorbed near calcium ions, which occupies most of the available adsorption sites, thereby preventing coordination between water molecules and calcium ions. Furthermore, the pathway from calcium ion dissolution to the 5-coordinated state—identified as the most favorable reaction path in terms of kinetics and thermodynamics—was analyzed (**Fig. 4c** and **4d**). Each step in the coordination transformation of calcium ions requires crossing two sequential energy barriers. For the initial state A, the process involves breaking two $Ca-O_s$ bonds from the surface, with free energy barriers of $\Delta A^\ddagger(A-B) = 4.64$ kJ/mol and $\Delta A^\ddagger(B-C) = 8.31$ kJ/mol. Subsequently, the calcium ion coordinates with an oxygen atom on the APTES silicon group to form state D, followed by the cleavage of another $Ca-O_s$ bond to reach state E. The free energy barriers for these steps are $\Delta A^\ddagger(C-D) = 17.32$ kJ/mol and $\Delta A^\ddagger(D-E) = 19.97$ kJ/mol.

It is important to emphasize that the final state of the WT-MetaD simulation aligns with the initial state (5,1), as this configuration exhibits the lowest energy and is thus the most stable on the free energy surface. As the coordination number decreases, the relative energies of states B, C, D, and E progressively increase compared to state A. This observation suggests that, in the presence of APTES, the calcium ion's stable state is more inclined toward a surface-bound position.

Fig. 4 Mechanisms of Ca_3SiO_5 dissolution under the influence of APTES. (a) Time-evolution of collective variables during the WT-MetaD simulation. (b) The two-dimensional free energy surface with variables of $CN(Ca-O_s)$ and $CN(Ca-O_{w+a})$ for the APTES-containing system. (c) The reaction coordinate representing the dissolution process, with the distinct transition states labeled as ‘TSn’ ($n=1, 2, 3, 4\dots$). (d) Corresponding snapshots illustrating the configuration evolution along the reaction pathway. The state numbers, coordinates on the FES, and the Helmholtz free energy values (in kJ/mol) relative to state A are indicated in the upper right corner. The red values upon the blue arrow denote

the free energy barriers (in kJ/mol), while the blue values beneath the arrows indicate the overall free energy changes between successive states (in kJ/mol). Atom color-coding is as follows: red for oxygen, yellow for silicon, white for hydrogen, cyan for calcium, gray for carbon, and pink for nitrogen. For clarity, water molecules are depicted in a transparent stick representation.

(Line 353, Page 20)

Obviously, the dissolution process of FES separated by Ca, which is completely exposed to solution, differs from FES adsorbed by APTES. The time-evolution of CVs (Fig. 5a) demonstrates that the highest tri-coordination with water is observed on the current free energy surface. However, both processes initiate from the same stable state, A(5,1). Depending on the sequence, three potential reaction pathways exist from state A to the adsorption of a water molecule, followed by the breaking of a Ca-O_s bond, leading to state B. From a thermodynamic perspective, it is more probable that these two steps occur simultaneously, as state B (4,2) exhibits a lower energy than both states (4,1) and (5,2), with the corresponding free energy barriers for this process being $\Delta A^+(A-B) = 12.10$ kJ/mol. Subsequently, Ca continues to adsorb a water molecule and break a Ca-O_s bond to reach the most stable C(3,3) state, which also represents the final state observed in MetaD simulations. For this step, there are two possible pathways. From a kinetic standpoint, the reaction is more likely to proceed along the B→C→D pathway (Fig. 5d), as the free energy barriers for this path— $\Delta A^+(B-C) = 10.77$ kJ/mol and $\Delta A^+(C-D) = 7.49$ kJ/mol—are lower than those of the B→D path ($\Delta A^+(B-D) = 16.52$ kJ/mol). To proceed to the E(2,3) state, Ca must overcome two consecutive free energy barriers of 10.12 kJ/mol.

Fig. 5 Mechanisms of Ca₃SiO₅ dissolution in aqueous solution. (a) Time-evolution of collective

variables during the WT-MetaD simulation. (b) The two-dimensional free energy surface with variables of $CN(\text{Ca-O}_s)$ and $CN(\text{Ca-O}_w)$. (c) The reaction coordinate representing the dissolution process, with the distinct transition states labeled as ‘TS $_n$ ’ ($n=1, 2, 3, 4\dots$). (d) Corresponding snapshots illustrating the configuration evolution along the reaction pathway. The state numbers, coordinates on the FES, and the Helmholtz free energy values (in kJ/mol) relative to state A are indicated in the upper right corner. The red values upon the blue arrow denote the free energy barriers (in kJ/mol), while the blue values beneath the arrows indicate the overall free energy changes between successive states (in kJ/mol).

In contrast to calcium ions that are fully exposed in an aqueous solution, the dissociative adsorption of APTES reduces the free energy barrier associated with initially breaking the two Ca-O_s bonds, thereby promoting the kinetic process. However, APTES inhibits the interaction between calcium ions and nearby water molecules, resulting in a consistent $CN(\text{Ca-O}_w)$ value of zero. From an energetic standpoint, the stable state of calcium is altered under the influence of silane, which leads to penta-coordination with the surface ($CN(\text{Ca-O}_s) = 5$). This is in contrast to the tri-coordinated calcium in an aqueous solution ($CN(\text{Ca-O}_s) = 3$), suggesting that calcium is more likely to stabilize at the surface following silane treatment. Additionally, the change in the free energy of the dissolution reaction further suggests that the spontaneity of the Ca dissolution process is influenced by silane. The transition from a decreasing to an increasing free energy change indicates that the dissolution reaction shifts from spontaneous to non-spontaneous, which provides a fundamental explanation for the inhibition of calcium dissolution by silane.

(Line 382, Page 22)

Comment 4: 1. 133-134 The initial distance was set to be 3 Å for physical adsorption and 2.5 Å for chemical adsorption. If possible, can you explain in few words why these distances and why they are different for physical and chemical adsorption.

Response: Thanks for your comment. The initial distances of 3 Å for physical adsorption and 2.5 Å for chemical adsorption were chosen based on the fundamental differences in the nature of these interactions.

Physical adsorption primarily involves weak van der Waals forces, which act at relatively larger separations due to their long-range nature. In this study, an initial distance of 2.5 Å was chosen to observe the interaction between the APTES silicon group and the silica tetrahedra on the Ca₃SiO₅ surface, dominated by hydrogen-bond interactions. As such, this interaction was initially classified as physical adsorption.

Chemical adsorption, in contrast, involves the formation of stronger covalent or ionic bonds, requiring closer proximity to enable orbital overlap or stronger electrostatic interactions. A distance of 3 Å, corresponding to typical Ca–O bond lengths observed in chemically bonded systems, was selected to simulate adsorption between surface calcium and APTES oxygen.

However, **both adsorption systems ultimately exhibit chemisorption through the formation of Ca–O bonds.** As a result, the classification in the manuscript was ambiguous, and we have revised it

accordingly.

The revised content can be found in the revised Manuscript:

Considering that SCAs adsorption is primarily governed by the silicon group, surface calcium and oxygen atoms (in silicate tetrahedra) were selected as the potential adsorption sites. The initial distance was set to 3 Å for the adsorption between surface calcium ions and APTES oxygen atoms, and 2.5 Å for the interaction between the APTES silicon group and the silica tetrahedra on the Ca_3SiO_5 surface.

(Line 140, Page 8)

Comment 5: I think that the acronym and DS is used without definition to qualify the dissociative adsorption and should be replaced by DA (see p.13 and check the whole manuscript). The same is true for MS and MA (molecular adsorption).

Response: Thanks for your comment. We have revised the “MS” to MA and check the full text to ensure the consistency of the shorthand.

The revised content could be found in the revised Manuscript:

Consequently, the DA adsorption mode is significantly more stable, making it more likely to take place when APTES interacts with the Ca_3SiO_5 surface.

(Line 288, Page 16)

Comment 6: l. 243-245: “Notably, the proton transfer that occurs in the DS process implies the occurrence of chemisorption, with more adsorption sites occupied and could impede the adsorption of water molecules nearby.”. Add spaces (“and could” + “molecules nearby”).

Response: Thanks for your comment. We have corrected these typos in the revised manuscript to make the logic of the analysis more coherent.

The revised content could be found in the revised Manuscript:

Notably, the proton transfer that occurs in the DA process implies the occurrence of chemisorption, with more adsorption sites occupied and could impede the adsorption of water molecules nearby. The formation of hydrogen bonds between the silanol group and Si tetrahedra on the Ca_3SiO_5 surface in the MS adsorption might offer chemical potential for the formation of Si-O-Si linkages.

(Line 274, Page 15)

Comment 7: l. 292-293 “In the DA condition, the Oa atom forms double Ca-Oa bonds with a -ICOHP value of 0.46, compared to 0.41 in the MA, indicating a stronger interaction.”. The 0.41 value is not corresponding to what is shown in Fig. 3d.

Response: Thanks for your comment. We have checked the relevant data and corrected the errors in the manuscript.

The revised content could be found in the revised Manuscript:

In the DA condition, the O_a atom forms double Ca- O_a bonds with a -ICOHP value of 0.462, compared to 393 in the MA, indicating a stronger interaction.

(Line 316, Page 17)

Comment 8: l. 377-378 “The lateral adsorption configuration of APTES molecules occupies multiple surface sites, thereby impeding the diffusion of water molecules to the calcium ion.”. This sentence is not directly related to the previous one and the reader may be a bit lost. Maybe add “Furthermore,”?

Response: Thanks for your comments. We have rewritten this part of analysis in the manuscript to make the logic more coherent.

The revised content could be found in the revised Manuscript:

However, APTES inhibits the interaction between calcium ions and nearby water molecules, resulting in a consistent $CN(Ca-O_w)$ value of zero. From an energetic standpoint, the stable state of calcium is altered under the influence of silane, which leads to penta-coordination with the surface ($CN(Ca-O_s) = 5$). This is in contrast to the tri-coordinated calcium in an aqueous solution ($CN(Ca-O_s) = 3$), suggesting that calcium is more likely to stabilize at the surface following silane treatment.

(Line 394, Page 22)

Reviewer #3

Comment 1: In the computational setup, why was the (111) surface of Ca_3SiO_5 chosen for the slab model? Are there other surfaces that might be relevant for this study?

Response: Thanks for your comment. In this work, we choose the (111) plane because it has lower surface energy at different degrees of cross-section, which has been proved by the previous DFT conclusions of Li et al.⁷ and MD simulations by Manzano et al.⁸ In general, surfaces with lower energy are easier to form in nature and therefore have a higher probability of distribution. Thus, we choose a low surface energy surface to represent a general case for study. As for other surfaces, the steric effect of silane on calcium dissolution would not be changed, while the free energy barriers may be altered. We believe that the surface we selected has been enough to qualitatively prove that the adsorption of silane inhibits the release of calcium ions from the surface of Ca_3SiO_5 . Since first-principles calculations are computationally costly, and the influence of the surface index on inhibitory effect of silane on Ca dissolution is not the primary research objective in this work, we will explore this in our future work.

Comment 2: The study focuses on APTES as the silane coupling agent. How might the results differ for other types of silanes commonly used in cement applications?

Response: Thanks for your comment. APTES was selected for this research due to its widespread use in engineering applications. Silane coupling agents generally consist of an organic component and an inorganic component ($-\text{SiO}_3$), with the silicon group primarily governing adsorption behavior on the surface of Ca_3SiO_5 . The adsorption of silicon groups on the surface effectively blocks the calcium ion adsorption sites, preventing contact with water molecules and thereby inhibiting further dissolution. Thus, various types of silane coupling agents may exhibit a similar molecular mechanism in inhibiting Ca dissolution. However, the associated free energy barriers are likely to differ.

Comment 3: In the experimental section, why was a 1% concentration of APTES chosen? How might varying concentrations affect the results?

Response: Thank you for your comment. The 1% dosage of APTES was selected based on relevant studies. Typically, the dosage of silane coupling agents ranges from 0.1% to 1% of the cement mass, with the exact amount adjusted based on performance test results for the material system, such as fluidity, mechanical properties, and durability. However, for qualitative studies—such as investigating the inhibition of hydration—a higher dosage, such as 1%, is often employed to produce a more pronounced retard effect. For instance, in Kong's study⁹, cement slurry treated with 1% silane by mass exhibited a significant delay in hydration.

Comment 4: The paper mentions that silane treatment may inhibit the nucleation and growth of C-S-H. Could the authors elaborate on the potential long-term effects of this inhibition on cement properties?

Response: Thanks for your comment. The inhibition effect on nucleation and growth of C-S-H could be well elucidated from the supplemented TEM snapshots in 60-, 120-minutes hydration. After silane treatment, the growth of C-S-H on the surface of Ca_3SiO_5 particles was significantly inhibited. In the blank group, distinct layered C-S-H structures were observed on the surface of Ca_3SiO_5 particles after 60 minutes of hydration. In contrast, similar layered structures were observed in the silane-treated group only after 120 minutes of hydration. Additionally, C-S-H particles were already present in the blank group at this time point.

The revised content could be found in the revised Manuscript:

Fig. 10c Representative TEM images at three distinct hydration times, where yellow dashed boxes highlight regions of dissolution inhibition on the surface of Ca_3SiO_5 particles in the silane-modified group.

(Line 526, Page 31)

Comment 5: The authors have not sufficiently contextualized their work within the existing body of literature on silane effects in cement hydration. For instance, the statement on lines 58-60 that “the inhibition mechanisms of SCAs on cement hydration remain elusive” overlooks significant prior work in this area. A more comprehensive review of existing studies on silane retardation mechanisms is necessary.

Response: Thanks for your comment. We contend that it is premature to conclusively assert that “the inhibition mechanisms of SCAs on cement hydration remain elusive”; rather, this topic requires more thorough investigation. Current understanding of the mechanism by which silane mixing inhibits hydration primarily revolves around two key concepts: firstly, the hydrophobic functional group of silane negatively impacts silicate polymerization and hydrogen bond formation; secondly, silane molecules may form complexes with calcium ions in solution. Both hypotheses offer plausible explanations for delayed hydration. However, we posit that these processes primarily influence the hydration phase following the induction period, as the concentration of calcium and silicate ions in solution approaches saturation during this stage. Conversely, the impact of silane during the induction period, specifically concerning its influence on clinker dissolution, remains unclear. Consequently, we revised this section to provide a more nuanced and progressive discussion, emphasizing the significance of silane’s effect on the dissolution process.

The revised content could be found in the revised Manuscript:

Moreover, the hydrophobic functional group of silane negatively impacts the silicate reaction and hydrogen bond formation, thereby influencing the nucleation of the C-S-H (calcium-silicate-hydrate) hydration product, which contributes to hydration delay. Xie et al. demonstrated that incorporating silane reduces the quantity and cumulative density of C-S-H, subsequently hindering the cement hydration process¹⁰. More recent findings indicate that silane hydrolysates can form complexes with calcium ions, potentially accounting for the delayed hydration¹¹. Despite these insights, there remain some gaps in the understanding of the mechanisms underlying silane-induced hydration delays. While the two aforementioned hypotheses provide plausible explanations, hydration is an inherently complex and prolonged process; thus, silane may exhibit different retarding mechanisms at various stages of hydration. Specifically, the mechanism of complexation and inhibition of C-S-H growth appears more relevant to the post-induction stage of hydration. However, the precise impact of silane on clinker dissolution during the induction stage warrants further exploration. Given that mineral dissolution occurs almost instantaneously, often within seconds, direct observation poses significant challenges. As such, atomic-scale simulations offer a promising approach for studying this phenomenon.

(Line 67, Page 4)

Comment 6: While the ab initio metadynamics simulations provide interesting insights, it’s not clear how these findings significantly advance our understanding beyond existing knowledge. The authors should clearly articulate how their results on calcium dissolution pathways (as shown in Figures 4 and 5) differ from or build upon previous studies.

Response: Thank you for your comment. To clarify the inhibitory effect of APTES molecules on Ca_3SiO_5 dissolution, we have re-analyzed the reaction pathways for both the silane-treated and blank

systems. Compared with the findings in Li's research⁶, we observed that while the free energy barrier for calcium ions detaching from the surface decreases in the presence of APTES, the ions tend to stabilize on the surface rather than undergo dissolution. This behavior is evident from the free energy surface (FES) landscape, which demonstrated from the consistency between the initial and final states, A(5,1). In addition, we have reanalyzed and revised the analytical section of the WT-MetaD simulation, as detailed below:

The negative adsorption energy of APTES on the Ca_3SiO_5 surface suggests that APTES preferentially adsorbs at reactive sites, potentially influencing the dissolution of Ca_3SiO_5 . To investigate these atomic-level interactions, *ab initio* WT-MetaD simulations were performed to examine the impact of APTES on the dissolution pathway of calcium ions. The two-dimensional free energy surface (FES) depicted in **Fig. 4b** illustrates the free energy barriers along the collective variables (CVs), with the corresponding numerical values represented by the color gradient. The free energy landscape for calcium ions reveals a range of minima distributed over a checkered pattern spanning 120 kJ/mol. These minima are associated with a reduction in the number of bonds to surface oxygen atoms of Ca_3SiO_5 and an increase in bonds with oxygen atoms from APTES. As the simulations progress, the free energy surface reveals an addition/elimination mechanism akin to the hydrolysis process observed in the aluminum phase of Al_2O_3 ⁵. Notably, even when the coordination number of water molecules and calcium ions is included as a set variable, the coordination bond between the oxygen atoms (O_w) of water molecules and calcium ions is not observed during the sampling process with the addition of Gaussian peaks (**Fig. 4a**).

This absence is attributed to both the hydrophobic effect of APTES and the silicon group adsorbed near calcium ions, which occupies most of the available adsorption sites, thereby preventing coordination between water molecules and calcium ions. Furthermore, the pathway from calcium ion dissolution to the 5-coordinated state—identified as the most favorable reaction path in terms of kinetics and thermodynamics—was analyzed (**Fig. 4c** and **4d**). Each step in the coordination transformation of calcium ions requires crossing two sequential energy barriers. For the initial state A, the process involves breaking two Ca– O_s bonds from the surface, with free energy barriers of $\Delta A^\ddagger(\text{A-B}) = 4.64$ kJ/mol and $\Delta A^\ddagger(\text{B-C}) = 8.31$ kJ/mol. Subsequently, the calcium ion coordinates with an oxygen atom on the APTES silicon group to form state D, followed by the cleavage of another Ca– O_s bond to reach state E. The free energy barriers for these steps are $\Delta A^\ddagger(\text{C-D}) = 17.32$ kJ/mol and $\Delta A^\ddagger(\text{D-E}) = 19.97$ kJ/mol.

It is important to emphasize that the final state of the WT-MetaD simulation aligns with the initial state (5,1), as this configuration exhibits the lowest energy and is thus the most stable on the free energy surface. As the coordination number decreases, the relative energies of states B, C, D, and E progressively increase compared to state A. This observation suggests that, in the presence of APTES, the calcium ion's stable state is more inclined toward a surface-bound position.

Fig. 4 Mechanisms of Ca_3SiO_5 dissolution under the influence of APTES. (a) Time-evolution of collective variables during the WT-MetaD simulation. (b) The two-dimensional free energy surface with variables of $\text{CN}(\text{Ca-O}_s)$ and $\text{CN}(\text{Ca-O}_{w+a})$ for the APTES-containing system. (c) The reaction coordinate representing the dissolution process, with the distinct transition states labeled as ‘TSn’ ($n=1, 2, 3, 4\dots$). (d) Corresponding snapshots illustrating the configuration evolution along the reaction pathway. The state numbers, coordinates on the FES, and the Helmholtz free energy values (in kJ/mol) relative to state A are indicated in the upper right corner. The red values upon the blue arrow denote the free energy barriers (in kJ/mol), while the blue values beneath the arrows indicate the overall free energy changes between successive states (in kJ/mol). Atom color-coding is as follows: red for oxygen, yellow for silicon, white for hydrogen, cyan for calcium, gray for carbon, and pink for nitrogen. For clarity, water molecules are depicted in a transparent stick representation.

(Line 353, Page 20)

Obviously, the dissolution process of FES separated by Ca, which is completely exposed to solution, differs from FES adsorbed by APTES. The time-evolution of CVs (Fig. 5a) demonstrates that the highest tri-coordination with water is observed on the current free energy surface. However, both processes initiate from the same stable state, A(5,1). Depending on the sequence, three potential reaction pathways exist from state A to the adsorption of a water molecule, followed by the breaking of a Ca-O_s bond, leading to state B. From a thermodynamic perspective, it is more probable that these two steps occur simultaneously, as state B (4,2) exhibits a lower energy than both states (4,1) and (5,2), with the corresponding free energy barriers for this process being $\Delta A^\ddagger(\text{A-B}) = 12.10$ kJ/mol. Subsequently, Ca continues to adsorb a water molecule and break a Ca-O_s bond to reach the most stable C(3,3) state, which also represents the final state observed in MetaD simulations. For this step, there are two possible pathways. From a kinetic standpoint, the reaction is more likely to proceed along

the B→C→D pathway (Fig. 5d), as the free energy barriers for this path— $\Delta A^*(B-C) = 10.77$ kJ/mol and $\Delta A^*(C-D) = 7.49$ kJ/mol—are lower than those of the B→D path ($\Delta A^*(B-D) = 16.52$ kJ/mol). To proceed to the E(2,3) state, Ca must overcome two consecutive free energy barriers of 10.12 kJ/mol.

Fig. 5 Mechanisms of Ca_3SiO_5 dissolution in aqueous solution. (a) Time-evolution of collective variables during the WT-MetaD simulation. (b) The two-dimensional free energy surface with variables of $\text{CN}(\text{Ca-O}_s)$ and $\text{CN}(\text{Ca-O}_w)$. (c) The reaction coordinate representing the dissolution process, with the distinct transition states labeled as ‘TS_n’ ($n=1, 2, 3, 4\dots$). (d) Corresponding snapshots illustrating the configuration evolution along the reaction pathway. The state numbers, coordinates on the FES, and the Helmholtz free energy values (in kJ/mol) relative to state A are indicated in the upper right corner. The red values upon the blue arrow denote the free energy barriers (in kJ/mol), while the blue values beneath the arrows indicate the overall free energy changes between successive states (in kJ/mol).

In contrast to calcium ions that are fully exposed in an aqueous solution, the dissociative adsorption of APTES reduces the free energy barrier associated with initially breaking the two Ca-O_s bonds, thereby promoting the kinetic process. However, APTES inhibits the interaction between calcium ions and nearby water molecules, resulting in a consistent $\text{CN}(\text{Ca-O}_w)$ value of zero. From an energetic standpoint, the stable state of calcium is altered under the influence of silane, which leads to penta-coordination with the surface ($\text{CN}(\text{Ca-O}_s) = 5$). This is in contrast to the tri-coordinated calcium in an aqueous solution ($\text{CN}(\text{Ca-O}_s) = 3$), suggesting that calcium is more likely to stabilize at the surface following silane treatment. Additionally, the change in the free energy of the dissolution reaction further suggests that the spontaneity of the Ca dissolution process is influenced by silane. The transition from a decreasing to an increasing free energy change indicates that the dissolution reaction shifts from spontaneous to non-spontaneous, which provides a fundamental explanation for the inhibition of calcium dissolution by silane.

(Line 382, Page 22)

Comment 7: The experimental methods section would benefit from more detail on sample preparation, particularly the mixing procedure for the Ca_3SiO_5 pastes with and without APTES.

Response: Thanks for your comment, we have rewritten this part to more clearly demonstrate our sample preparation procedures:

In the silane treatment Ca_3SiO_5 paste, Ca_3SiO_5 powder was weighed and subsequently mixed with the APTES solution while l/p ratio remained unchanged with only water in the blank group. The resulting paste underwent rapid magnetic stirring at 100 rpm for 40 seconds to ensure uniform mixing of the solution and the powder, followed by curing of the paste.

(Line 210, Page 12)

Comment 8: The paper lacks a comprehensive mechanistic model that integrates both the observed dissolution inhibition and the known hydrolysis/condensation reactions of silanes. This is necessary to provide a full picture of silane effects in cement systems.

Response: Thanks for your comment. We fully understand your concern about the hydrolysis/condensation reactions of silanes influencing the dissolution inhibition effect. The hydrolysis of the silane monomer facilitates its adsorption onto the surface of Ca_3SiO_5 , as the $-\text{SiOH}$ group provides additional sites for the formation of Ca-O and hydrogen bonds. Consequently, silane hydrolysis aids in inhibiting the dissolution of calcium ions from the surface. For the condensation, two types of silane condensation reactions are known: self-condensation and condensation with the hydroxyl groups on the substrate surface^{12,13}. Silane polymers formed via self-condensation may increase steric hindrance, further inhibiting dissolution. However, this process also reduces the available numbers of $-\text{SiOH}$ groups, which may diminish the adsorption strength on the Ca_3SiO_5 surface, a hypothesis that warrants further investigation. On the other hand, the condensation reaction between silane and the substrate surface results in the formation of Si-O-Si covalent bonds, thereby enhancing surface adhesion and contributing to the inhibition of dissolution.

Furthermore, the hydrolysis and condensation reactions of silanes may influence their dissolution-inhibiting effect. Hydrolysis of the silane monomer promotes its adsorption onto the surface of Ca_3SiO_5 , as the $-\text{SiOH}$ group provides additional sites for the formation of Ca-O and hydrogen bonds. As a result, silane hydrolysis plays a crucial role in inhibiting the dissolution of calcium ions from the surface. In terms of condensation, silane polymers formed through self-condensation may increase steric hindrance, further preventing dissolution. However, this process also reduces the number of available $-\text{SiOH}$ groups, which could potentially weaken adsorption strength on the Ca_3SiO_5 surface—an aspect

that warrants further exploration. In contrast, condensation between silane and the substrate surface leads to the formation of Si-O-Si covalent bonds, thereby enhancing surface adhesion and contributing to the inhibition of dissolution.

(Line 552, Page 32)

Comment 9: Consider adding a schematic diagram illustrating the proposed mechanism of silane inhibition on Ca_3SiO_5 dissolution. This would help readers visualize the key findings.

Response: Thank you for your comment. A schematic diagram has been attached, which appropriately illustrates our novel findings.

Comment 10: The conclusion could be strengthened by explicitly stating the broader implications of this research for the development of organic-inorganic cement composites.

Response: Thanks for your comment. We have strengthened the conclusion from the following aspects: Firstly, by elucidating the molecular mechanisms underlying silane's interaction with primary clinker phase, this study provides critical insights into the hydration behavior of organic-modified cement systems. Secondly, The demonstrated ability of silane to stabilize calcium ions and hinder dissolution highlights its potential to enhance the durability, mechanical performance, and chemical resistance of cement-based materials. This work establishes a theoretical foundation for designing next-generation organic-inorganic composites with improved performance characteristics, paving the way for more sustainable and functional construction materials.

The revised content could be found in the revised Manuscript:

By elucidating the molecular mechanisms underlying silane's interaction with cementitious phases, this study provides critical insights into the hydration behavior of organic-modified cement systems.

The demonstrated ability of silane to stabilize calcium ions and hinder dissolution highlights its potential to enhance the durability, mechanical performance, and chemical resistance of cement-based materials. This work establishes a theoretical foundation for designing next-generation organic - inorganic composites with improved performance characteristics, paving the way for more sustainable and functional construction materials.

(Line 562, Page 33)

Comment 11: For the ICP-OES analysis (Fig. 9), provide more data points, especially in the early stages of hydration (0-20 minutes) to better capture the initial dissolution kinetics.

Response: Further ICP tests were conducted to determine the calcium ion concentration of Ca_3SiO_5 within 20 minutes of hydration, including hydration time points 3, 6, 9, 12, 15, 18, and the results are shown in Fig. 8c.

The results revealed that additional silane significantly reduced both the dissolution and consumption rate of calcium ion. Notably, the calcium ion concentration in the silane-treatment group was substantially lower than that in the blank group during the initial 15 minutes of hydration.

Fig. 8c Concentration profile of dissolved calcium ions over the initial 20-minute hydration period.

These data well reflect the kinetics of inhibition of initial calcium dissolution by silane, which can be found in the revised manuscript.

(Line 475, Page 27)

Comment 12: In the XPS analysis (Fig. 10), include quantitative data on peak shifts and intensities. A table summarizing these changes across different hydration times would be valuable.

Response: Thanks for your comment. We have supplemented the data of the peak and displacement of XPS in the manuscript.

The supplemented table could be found in the revised Manuscript:

Table 3 Quantitative data of XPS including coordinates and intensities of diffraction peaks.

Parameters	Binding energy (eV)				Intensity (cps)			
	Silane-treatment		Blank		Silane-treatment		Blank	
Orbitals	Ca2p	Si2p	Ca2p	Si2p	Ca2p	Si2p	Ca2p	Si2p
10 min	347.2	101.7	346.7	101.0	29879.2	2852.26	43981.3	5444.31
30 min	347.0	101.2	346.7	101.1	34605.1	3887.22	37633.9	4723.17
60 min	346.9	101.2	346.7	101.1	35542.3	4008.84	45077.5	5652.65
120 min	346.8	101.2	346.8	101.3	39105.9	4773.82	31430.1	4728.02

(Line 501, Page 29)

Comment 13: For the heat flow measurements (Fig. 8), extend the monitoring period beyond 3 days to capture any long-term effects of silane on hydration kinetics.

Response: The heat flow analysis indicated that silane treatment had no significant long-term impact on the heat release rate of Ca_3SiO_5 slurry after the system reached the deceleration stage, at least within the 4-day hydration period. This finding demonstrates that the hydration delay caused by silane primarily affects the dissolution, induction, and acceleration stages.

The supplemented content could be found in the revised manuscript:

Fig. 8d Heat flow and cumulative heat release curves of Ca_3SiO_5 during the hydration process.

The revised content could be found on Line X of the revised manuscript.

(Line 475, Page 27)

Comment 14: Include Rietveld analysis of XRD data to quantify the phase composition changes during hydration, which would complement the XPS and calorimetry results.

Response: Thank you for your comment. We fully agree that quantitative X-ray diffraction (QXRD) analysis provides an intuitive understanding of phase composition changes. Accordingly, we

conducted QXRD tests on Ca_3SiO_5 samples after 10, 30, 60, and 120 minutes of hydration to analyze the compositional differences introduced by silane treatment. The presented results show that the continuous dissolution of Ca_3SiO_5 was observed throughout the hydration induction period. Silane treatment notably reduced the hydration degree of Ca_3SiO_5 , resulting in a higher relative content compared to the blank group at the same time points. This analysis supports the results that silane treatment reduces the dissolution rate of Ca_3SiO_5 , thus delaying its early hydration.

The supplemented content could be found in the revised manuscript:

Fig. 10a and 10b Phase Structure Analysis During Early Hydration Stages. (a) QXRD curves at various hydration intervals, with standard PDF cards for Ca_3SiO_5 and ZnO highlighted. (b) Phase content derived from XRD fitting analysis, where the R-value for each group is maintained below 12%. “B” denotes the pure water group, and “S” represents the silane-modified group, with phase content expressed as weight percentage (wt%) of the sample.

QXRD curve showed that the characteristic peak of calcium hydroxide disappearing, even after 120 minutes of hydration corresponding to the accelerated dissolution stage. During this stage, the ion concentration in the solution had not yet reached the saturation threshold necessary for precipitation. Furthermore, phase components at each hydration interval of both groups were quantitatively calculated using fitting analysis to compare the mass content of Ca_3SiO_5 consumption. As depicted in Fig. 10b, the continuous dissolution of Ca_3SiO_5 was observed throughout the hydration induction period. Silane treatment notably reduced the hydration degree of Ca_3SiO_5 , resulting in a higher relative content of Ca_3SiO_5 compared to the untreated (blank) group at the same time points.

(Line 526, Page 31)

Comment 15: Provide more details on the AIMD simulations, including total simulation time and time step used. This information is crucial for assessing the reliability of the computational results.

Response: Thanks for your comment, the total simulation time and used time step now have been supplemented to the manuscript.

The supplemented content could be found in the revised Manuscript:

The AIMD time step was initially set at 0.5 fs for relaxation, while a time step of 1.0 fs was used for WT-MetaD simulations, final structural analysis, and subsequent MetaD simulations. The final structural analysis simulations are with the replacement of hydrogen by deuterium to accelerate the structural evolution without energy drifts. Total AIMD simulation time was presented in Table 1:

Table 1 Production time of AIMD simulations

Production details	Time
Surface relaxation after explicit solution introduced	10 ps
Dissolution pathway of Ca under APTES influence	66 ps
Dissolution pathway of Ca in explicit aqueous solution	63 ps
Configuration analysis of final state relaxation from WT-MetaD	10 ps for each system

(Line 160, Page 9)

Comment 16: Include additional characterization techniques such as High resolution TEM to visualize the microstructural changes induced by silane treatment, particularly at the Ca_3SiO_5 -silane interface, to provide a quantitative comparison between their observed retardation effects and those reported in previous literature.

Response: Thanks for your comment. TEM experiments were conducted at 10-, 60-, and 120-minutes hydration of Ca_3SiO_5 to visualize the microstructural changes, and the representative snapshots are shown in Fig. 10c. After 10 minutes of hydration, a discernible precipitation layer had formed on the surface of Ca_3SiO_5 particles in the blank group. In contrast, silane-treated particles exhibited a thinner precipitation layer, and in some regions, precipitation was absent. Additionally, hydration gradually disrupted the Ca_3SiO_5 particle lattice, forming disordered, layered C-S-H (calcium silicate hydrate) products on particle surfaces in the blank group within 60 minutes. This transformation was delayed in the silane-treated group, appearing only after 120 minutes, at this point, nearly complete dissolution of Ca_3SiO_5 particles was observed in the blank group. These snapshots effectively validate the primary finding of this study: the limited dissolution of Ca_3SiO_5 particle surfaces following silane treatment. This phenomenon not only delays the hydration process.

The TEM analysis could be found in the revised Manuscript:

Fig. 10c Representative TEM images at three distinct hydration times, where yellow dashed boxes highlight regions of dissolution inhibition on the surface of Ca_3SiO_5 particles in the silane-modified group.

TEM images further confirmed the reduced dissolution rate induced by silane treatment in silane treated Ca_3SiO_5 paste. After 10 minutes of hydration, a discernible precipitation layer was formed and coated on the surface of Ca_3SiO_5 particles in the blank group (Layer A). In contrast, silane-treated Ca_3SiO_5 exhibited a thinner precipitation layer (Layer B), and in some regions, precipitation layer was entirely absent. Additionally, hydration gradually disrupted the Ca_3SiO_5 particle lattice, forming disordered, layered C-S-H (calcium silicate hydrate) products on particle surfaces in the blank group within 60 minutes. However, this transformation was postponed in the silane-treated group, appearing until 120 minutes, when nearly complete dissolution of Ca_3SiO_5 particles was observed in the blank group.

(Line 526, Page 31)

Comment 17: Consider performing isothermal titration calorimetry (ITC) experiments to directly measure the binding energies between silane and Ca_3SiO_5 surfaces, which could validate the computational adsorption energy calculations.

Response: Thank you for your insightful comment. We have performed isothermal titration calorimetry (ITC) analysis to directly measure the entropy (ΔS) and enthalpy (ΔH) changes associated with the chemical reaction between APTES and the Ca_3SiO_5 particle surface. Based on these results, the Gibbs free energy (ΔG) was calculated using two approaches: (1) directly from the original data and (2) from the fitted curve slope.

For the former, ΔG was calculated using the equation $\Delta G = \Delta H - T\Delta S$, yielding a value of -18.2632 kJ/mol. For the latter, ΔG was determined using the equation $\Delta G = -RT\ln K$, where K was obtained by fitting after subtracting the last five stable data points, resulting in $\Delta G = -19.1466$ kJ/mol. Despite these calculations, a significant discrepancy remains between the experimental Gibbs free energy values and the adsorption energy derived from the simulation.

This difference arises because adsorption energy represents the energy released or absorbed when molecules or ions are adsorbed onto a solid surface. It encompasses chemical bonding, interfacial recombination, and electron transfer energy (for chemisorption) and primarily describes the strength of interactions between molecules and surfaces. In contrast, ΔG is a thermodynamic state function influenced by internal energy, volume work, entropy, and temperature, serving primarily to predict the spontaneity and directionality of chemical reactions, phase transitions, and other processes. As such, a mismatch between the two energy metrics is inevitable. We did not consider this value to indicate any substantive issues; therefore, the ITC results were not included in the manuscript.

In this study, the calculated negative Gibbs free energy between Ca_3SiO_5 and APTES indicates that the adsorption process is spontaneous. Furthermore, it is worth noting that in ITC experiments, the apparent free energy may also reflect the coupling of multiple adsorption reactions, which could further influence the observed ΔG .

Experiment conditions:

ITC experiments were conducted using the Malvern MicroCal VP-ITC (UK). The Ca₃SiO₅ powder was first mixed with anhydrous ethanol (99.99%, Aladdin) at a concentration of 0.6667 mM, and the APTES monomer was dissolved in the anhydrous ethanol at a concentration of 5 mM. The injection sequence consisted of an initial injection of 1 µL followed by injections of 2 µL with an interval of 180 s between each. The duration of each injection was 4 s. The initial injection of 1 µL was used to prevent artifacts arising from the filling of the syringe and was not used in data analysis. The contents of the sample cell were stirred continuously at a rate of 400 rpm. The reference power was set to 22 µcal/s. The experiments were performed at 25 °C.

Thank you again for your time and profound comments on our manuscript.

Best regards,

All authors

1. Manzano, H., Durgun, E., López-Arbeloa, I. & Grossman, J. C. Insight on tricalcium silicate hydration and dissolution mechanism from molecular simulations. *ACS Appl. Mater. Interfaces* **7**, 14726–14733 (2015).
2. Xu, X. *et al.* The initial stages of cement hydration at the molecular level. *Nat. Commun.* **15**, 2731 (2024).
3. Jalilehvand, F. *et al.* Hydration of the calcium ion. An EXAFS, large-angle X-ray scattering, and molecular dynamics simulation study. *J. Am. Chem. Soc.* **123**, 431–441 (2001).
4. Zhang, W. *et al.* Study on unsaturated transport of cement-based silane sol coating materials. *Coatings* **9**, 427 (2019).
5. Réocreux, R. *et al.* Reactivity of shape-controlled crystals and metadynamics simulations locate the weak spots of alumina in water. *Nat. Commun.* **10**, 3139 (2019).
6. Li, Y., Pan, H., Liu, Q., Ming, X. & Li, Z. Ab initio mechanism revealing for tricalcium silicate dissolution. *Nat. Commun.* **13**, 1253 (2022).
7. Li, Y. *et al.* Insight into adsorption mechanism of water on tricalcium silicate from first-principles calculations. *Cem. Concr. Res.* **152**, 106684 (2022).
8. Durgun, E., Manzano, H., Kumar, P. V & Grossman, J. C. The characterization, stability, and reactivity of synthetic calcium silicate surfaces from first principles. *J. Phys. Chem. C* **118**, 15214–15219 (2014).
9. Kong, X.-M., Liu, H., Lu, Z.-B. & Wang, D.-M. The influence of silanes on hydration and strength development of cementitious systems. *Cem. Concr. Res.* **67**, 168–178 (2015).
10. Xie, M., Zhong, Y., Li, Z., Lei, F. & Jiang, Z. Study on alkylsilane-incorporated cement composites: Hydration mechanism and mechanical properties effects. *Cem. Concr. Compos.* **122**, 104161 (2021).
11. Yang, J., Zuo, W. & She, W. Towards a further understanding of cement hydration at the early-age stage in the presence of hydrophobic silane IBTEO. *Cem. Concr. Compos.* 105712 (2024).
12. Stewart, A., Schlosser, B. & Douglas, E. P. Surface modification of cured cement pastes by silane coupling agents. *ACS Appl. Mater. Interfaces* **5**, 1218–1225 (2013).
13. Yu, J., Zheng, H., Hou, D., Zhang, J. & Xu, W. Silane coupling agent modification treatment to improve the properties of rubber--cement composites. *ACS Sustain. Chem. Eng.* **9**, 12899–12911 (2021).